



# Biomarker characterization of the North Water Polynya, Baffin Bay: Implications for local sea ice and temperature proxies

David J. Harning[1], Brooke Holman[1], Lineke Woelders[1], Anne E. Jennings[1], Julio Sepúlveda[1, 2]

[1]Institute of Arctic and Alpine Research, University of Colorado, Boulder, USA

[2]Department of Geological Sciences, University of Colorado, Boulder, USA

*Correspondence to*: David J. Harning (david.harning@colorado.edu)

**Abstract.** The North Water Polynya (NOW, Greenlandic Inuit: *Pikialasorsuaq*), Baffin Bay, is the largest polynya and one of the most productive regions in the Arctic. This area of thin to absent sea ice is a critical moisture source for local ice sheet sustenance and coupled with the inflow of nutrient-rich Arctic Surface Water, supports a diverse community of Arctic fauna

and indigenous people. Although paleoceanographic records can provide critical insight into the NOW's past behavior, it is critical that we fully understand the modern functionality of the paleoceanographic proxies beforehand. In this study, we analyzed lipid biomarkers, including algal highly-branched isoprenoids and sterols for sea ice extent and pelagic productivity, and algal alkenones and archaeal GDGTs for ocean temperature, in a suite of modern surface sediment samples from within and around the NOW. Our data show that all highly-branched isoprenoids exhibit strong correlations with each other and show

highest concentrations within the NOW, which suggests a spring/autumn sea ice diatom source rather than a combination of sea ice and open water diatoms as seen elsewhere in the Arctic. Sterols are also highly concentrated in the NOW and exhibit an order of magnitude higher concentration here compared to sites south of the NOW, consistent with the order of magnitude higher primary productivity observed within the NOW relative to surrounding waters in spring/summer months. Finally, our temperature calibrations for alkenones, GDGTs and OH-GDGTs reduce the uncertainty present in global temperature

calibrations, but also identify some additional variables that may be important in controlling their local distribution, such as salinity, nutrients, and dissolved oxygen. Collectively, our datasets provide new insight into the utility of these lipid biomarker proxies in high-latitude settings and will help provide a refined perspective on the Holocene development of the NOW with their application in downcore reconstructions.

## 1 Introduction

Arctic and Antarctic polynyas are key sites for deep water formation (Kuhlbrodt et al., bvc2017), moisture sources for adjacent ice sheets (Smith et al., 2010), and enhanced productivity that can sequester atmospheric $CO_2$ (Arrigo and van Dijken, 2004). As global temperatures continue to climb, further reductions in sea ice are projected along Arctic coastlines (Barnhart et al., 2015; Onarheim et al., 2018), calling the future status of polynyas into question. Polynyas will cease to exist where seasonal sea ice vanishes or transitions to marginal sea ice zones, which will both result in cascading negative effects on regional and



global environments (Meredith et al., 2019; Moore et al., 2021). One way to alleviate some uncertainty about the future status of polynyas is by reconstructing changes in sea ice extent and productivity in the recent geologic past in order to understand how polynyas have responded to past climate change. In this light, the Holocene instability of coastal polynyas has recently been shown in the Barents Sea (Knies et al., 2018), East Greenland (Syring et al., 2020) and northern Baffin Bay (Jackson et al., 2021) using lipid biomarker climate proxies, such as IP$_{25}$ (sea ice) and sterols (open water productivity). However,

understanding the modern distributions of biomarkers and the influencing environmental factors within dynamic Arctic regions is critical before making down-core interpretations (e.g., Smik and Belt, 2017; Belt et al., 2019; Kolling et al., 2020).

Here, we focus on the North Water Polynya (NOW, Greenlandic Inuit: *Pikialasorsuaq*), which is the largest polynya (85,000 km$^2$) and one of the most productive regions in the Arctic (Barber et al., 2001; Klein et al., 2002). The latent heat NOW forms when an ice bridge consolidates at the head of Smith Sound, which restricts the passage of Arctic ice floes but

allows throughflow of nutrient-rich Arctic Surface Water (ASW) into Baffin Bay through Lancaster and Jones Sounds (Fig. 1, Melling et al., 2001). This throughflow and open water fuels high local productivity, which supports a diverse community of Arctic fauna, including higher-trophic level seabirds and marine mammals (Stirling, 1980; Tremblay et al., 2002a). Due to the rich biodiversity, the region has also supported intermittent human occupation for at least 4000 years (Dorset, 500 BCE to 1500 CE; Thule, 1000 to 1600 CE, Schlederman, 1980), with modern Inuit inhabitants continuing to rely on the NOW for their

food security and subsistence economy today (Hastrup et al., 2018). Given that the sea-ice floes fail to block Smith Sound as reliably as they once did (Moore et al., 2021), the NOW is becoming geographically and seasonally less defined (Ryan and Münchow, 2017).

In this study, we aim to characterize the NOW through the distribution of lipid biomarkers archived in marine seafloor surface sediments taken in 2008 and 2017 that encompass its modern area in Baffin Bay. We focus on different lipid classes

that inform us about seasonal sea ice extent, surface productivity and ocean temperature. Our assessment of these biomarker proxies against modern instrumental data provides a key baseline for interpreting the presence and extent of the NOW in the geologic past (Georgiadis et al., 2020; Jackson et al., 2021). On a broader scale, our work is also critical for the community's general understanding of these lipids' environmental relationships at high northern latitudes where datasets are currently sparse (e.g., Tierney and Tingley, 2014, 2018). For example, existing biomarker temperature calibrations are often global in scale and

feature high uncertainties at the low end of the temperature spectrum (Müller et al., 1998; Kim et al., 2010, 2012). However, by generating regional calibrations, uncertainty can be substantially reduced by isolating the distinct regional characteristics of the environment (Tierney and Tingley, 2014, 2018; Harning et al., 2019).

## 2. Study Area

Ocean circulation in Baffin Bay is cyclonic, involving the northward flowing West Greenland Current (WGC) and the

southward flowing Baffin Current (BC) (Fig. 1). The warm and saline WGC carries a mixture of Atlantic Water from the Irminger Current and Polar Water from the East Greenland Current, whereas the BC is comprised of low salinity Arctic surface water (ASW) that enters Baffin Bay from the Arctic Ocean through the Canadian Arctic Archipelago (CAA) channels. The



ASW is modified by mixing with terrestrial-derived freshwater and by sea-ice processes en route to Baffin Bay (Tang et al., 2004; Münchow et al., 2006, 2015, Azetsu-Scott et al., 2010). The present-day depths of the CAA channels govern the composition of inflowing ASW (Jones et al., 2003); Nares Strait has a sill depth of 220 m that allows passage of both the Polar Mixed Layer (containing high-nutrient Pacific Water from Bering Strait) and some of the halocline layer that has been mixed with the underlying Atlantic layer of the Arctic Ocean (Azetsu-Scott et al., 2010). Lancaster and Jones Sounds have shallower sill depths that exclude all but the most carbonate-undersaturated Polar Mixed Layer (Azetsu-Scott et al., 2010). These Arctic outflows join the BC and form the upper 100 to 300 m of surface water in Baffin Bay, except where the WGC dominates in the southeast (Tang et al., 2004).

Sea ice covers nearly all of Baffin Bay in winter, except in the southeast due to the warmth and salinity of the WGC (Fig. 1). Sea ice begins to form in September and reaches maximum coverage in March and is thickest along the Baffin Island coast where the ASW flow is concentrated (Fig. 1, Tang et al., 2004). In contrast, the NOW has anomalously low concentrations of thin sea ice, even during winter months. Consolidation of an ice arch at the head of Smith Sound initiates the formation of the polynya, which is further stimulated by northerly winds and currents that remove newly formed sea ice (Ingram et al., 2002; Bi et al., 2019), and sensible heat from WGC upwelling on the Greenland side (Melling et al., 2001; Ingram et al., 2002). Baffin Bay sea ice concentrations decrease between April and August; beginning in the NOW region before propagating southward and creating a generally ice-free Baffin Bay by June (Bi et al., 2019). The Pacific Water, a major component of the ASW, has twice the nitrogen and phosphorus and seven times the silica of Atlantic Water (Jones et al., 2003). The high nutrient content of incoming ASW, along with higher light levels and stratification in the NOW, fuels high seasonal phytoplankton productivity (Lewis et al., 1996; Ingram et al., 2002; Tremblay et al., 2002a). Productivity is an order of magnitude higher in the NOW than in adjacent areas of Baffin Bay, making it one of the most important areas for new production in the Arctic (Tremblay et al., 2002a).

## 3. Background on lipid biomarkers

Highly branched isoprenoids (HBIs) are unsaturated hydrocarbons biosynthesized by a narrow range of marine diatoms (see review by Belt, 2018). The mono-unsaturated HBI termed $IP_{25}$, first discovered in Canadian Arctic sea ice (Belt et al., 2007), has developed into an important seasonal sea ice proxy due to its accumulation during spring blooms (Brown et al., 2011; Limoges et al., 2018; Amiraux et al., 2019) of Arctic sea ice diatoms (Brown et al., 2014). Based on a distinctly heavy stable carbon isotopic composition, in addition to similar concentration profiles to $IP_{25}$ across Arctic marine surface sediment, the di-unsaturated HBI II also likely has an Arctic sea ice diatom source (Belt et al., 2008; Cabedo-Sanz et al., 2013; Brown et al., 2014; Limoges et al., 2018). $IP_{25}$ and HBI II below the limit of detection have often been interpreted as either reflecting a lack of seasonal sea ice cover or permanent and thick sea ice that blocks sunlight penetration needed for sea ice diatom photosynthesis. However, this is likely an oversimplification of a broader range of scenarios that result in absent $IP_{25}$ (Belt, 2018). Other HBIs, such as the tri-unsaturated isomers HBI III and IV, have been attributed to biosynthesis by open-water phytoplankton (Belt et al., 2000, 2008, 2015, 2017; Rowland et al., 2001), which may help in differentiating between



open water or thick sea ice conditions inferred from $IP_{25}$ and HBI II in the Arctic (Cabedo-Sanz et al., 2013; Smik et al., 2016; Köseoğlu et al., 2018). Along with certain sterols (see following paragraph), the $PIP_{25}$ index has been developed for semi-quantitative sea-ice reconstructions (e.g., Müller et al., 2011):

$$PIP_{25} = \frac{IP_{25}}{IP_{25} + (phytoplankton\ biomarker\ x\ c)},$$  (1)

where the balance factor c is a ratio of mean $IP_{25}$ and mean phytoplankton biomarker concentrations. Although in some regions high concentrations of HBI III have also been associated with highly productive marginal ice zones (Barents Sea, Belt et al., 2019), marine fronts (North Iceland Shelf, Harning et al., 2020) and sea ice (Amiraux et al., 2019, 2020; Koch et al., 2020) that may obscure $PIP_{25}$-derived indices, recent compilations of Arctic surface sediments show that $PIP_{25}$-based indices broadly correlate with spring and autumn sea ice concentrations (Xiao et al., 2015; Kolling et al., 2020).

Sterols are ubiquitous components in eukaryotic organisms (Volkman,1986), and similar to HBI III and IV, have become common complimentary biomarkers in $IP_{25}$ and $PIP_{25}$ datasets. Although these biomarkers have often been attributed to specific sources, such as pelagic phytoplankton (brassicasterol, e.g., Navarro-Rodriguez et al., 2013), dinoflagellates (dinosterol, e.g., Boon et al., 1979) and vascular plants (campesterol and β-sitosterol, e.g., Huang and Meinschein, 1976), they are now known to derive from variable sources that complicate their source specificity. For example, brassicasterol and

dinosterol are also found in sea ice (Nichols et al., 1990; Belt et al., 2013, 2018) and pennate diatoms (e.g., Volkman et al., 1993; Rampen et al., 2010), and campesterol and β-sitosterol can be produced by diatoms as well (Belt et al., 2013, 2018; Rampen et al., 2010). Hence, the abundance of these sterols within marine sedimentary records are more broadly reflective of general marine productivity (e.g., Köseoğlu et al., 2019). In the Arctic where terrestrial biomass is low, we assume that the contribution of terrestrial-derived campesterol and β-sitosterol is minimal compared to that produced in the ocean.

Alkenones are long-chain ketones produced by haptophyte algae in the photic zone (Volkman et al., 1980; Conte et al., 1995). Culture and empirical global core top studies have demonstrated that the number of unsaturations reflected in the $U^{K}_{37}$ indices (e.g., $U^{K}_{37}$ and $U^{K'}_{37}$) vary in response to growth temperature and these indices are strongly correlated with annual sea surface temperature (SST) (Brassell et al., 1986; Prahl and Wakeham, 1987; Müller et al., 1998). In addition to temperature, $U^{K}_{37}$ indices may also be affected by variations in nutrient concentrations, light limitation, diagenesis, and algal source (Hoefs

et al., 1998; Gong and Hollander, 1999; Prahl et al., 2003; Rontani et al., 2013; Wang et al., 2021). Previous studies have also questioned the applicability of alkenone-derived temperature proxies at high latitudes, due to the nonlinearity of the $U^{K}_{37}$ index and SST relationship at low temperatures (<6 ∘C) and the high and erratic abundance of the $C_{37:4}$ alkenone (e.g., Sikes and Volkman, 1993; Rosell-Melé et al., 1994; Rosell-Melé, 1998; Rosell-Melé and Comes, 1999; Conte et al., 2006). The relative abundance of $C_{37:4}$ also exhibits a strong correlation with surface salinities in polar and sub-polar regions (Rosell-Melé, 1998;

Sicre et al., 2002; Harada et al., 2003; Bendle et al., 2005; Blanz et al., 2005), although a recent combination of DNA sequencing, culturing and alkenone analyses demonstrates that, at least for the Canadian Arctic, elevated $C_{37:4}$ abundance is produced by the sympagic Group 2i Isochrysidales lineage rather than the more widespread pelagic coccolithophores (*Emiliania huxleyi* and *Gephyrocapsa oceanica*) alkenone producers (Wang et al., 2021).



Isoprenoid GDGTs are cell membrane lipids biosynthesized by marine ammonia oxidizing Thaumarchaeota

(Schouten et al., 2002; Könneke et al., 2005; Pitcher et al., 2011; Besseling et al., 2020). The degree of cyclization reflected in various $TEX_{86}$ indices in global surface sediment datasets has been interpreted to represent a physiological response of Thaumarchaeota to *in situ* temperature variability (Schouten et al., 2002, 2013; Kim et al., 2010, 2012). Evidence from a latitudinal transect in the western Atlantic Ocean demonstrates that GDGTs are most likely produced and exported to the seafloor from 80–250 m water depth (Hurley et al., 2018), which compares well to archaea abundance maxima at 200 m water

depths in the Pacific Ocean (Karner et al., 2001). Considering that Thaumarchaeota are chemolithoautotrophs that perform ammonia oxidation (conversion of ammonia to nitrite), they are typically more abundant around the primary nitrite maximum near the base of the photic zone (Church et al., 2010; Francis et al., 2005; Hurley et al., 2018) and are most productive when there is minimized phytoplanktic competition over ammonia (Schouten et al., 2013). In the higher latitudes, the latter occurs during the less productive dark winter months when photosynthesis for sea surface species is inhibited, which may explain the

seasonal winter temperature bias of GDGTs observed in this latitudinal band (Herfort et al., 2006; Rueda et al., 2009; Rodrigo-Gámiz et al., 2015; Harning et al., 2019). Although the temperature relationship of $TEX_{86}$-based indices deviates from the linear global temperature calibrations and features higher uncertainty at the cold end of the spectrum (Schouten et al., 2002; Kim et al., 2010, 2012), regional calibrations have proven useful for reducing estimate uncertainty (Harning et al., 2019) that may be partially attributed to changes in community composition (e.g., Elling et al., 2017). Moreover, new indices based on

hydroxylated isoprenoid GDGTs (OH-GDGTs, e.g., RI-OH and RI-OH' indices, Lü et al., 2015) produced by planktic Thaumarchaeota (Elling et al., 2017; Bale et al., 2019) have been suggested to improve upon $TEX_{86}$-based proxies in the polar oceans (Fietz et al., 2013, 2020; Huguet et al., 2013) as the addition of one hydroxyl group may further reduce membrane rigidity at lower temperatures (Huguet et al., 2017). In addition to temperature, recent studies have shown that several other environmental and geochemical factors can influence the degree of GDGT cyclization, such as growth phase (Elling et al.,

2014), ammonia oxidation rates (Hurley et al., 2016), and oxygen concentrations (Qin et al., 2015), but not likely salinity (Wuchter et al., 2004, 2005; Elling et al., 2015). Although the effects of these environmental parameters on OH-GDGT cyclization has not been as rigorously tested, emerging evidence suggests that salinity, sea ice, seasonality, and terrestrial input may be complicating factors in some oceanic settings (Fietz et al., 2013; Kang et al., 2017; Lü et al., 2019; Park et al., 2019; Wei et al., 2020).

## 155 4. Methods

### 4.1. Marine surface sediment samples

The marine surface sediment samples studied here (*n*=13) are box core and trigger core tops collected during the 2008 CSS *Hudson* and 2017 CSS *Amundsen* research cruises, and range in water depths from 267 to 2373 m bsl (Table 1). As each sample integrates the upper 1 to 2 centimeters of sediment, they may reflect several decades to centuries of time depending

upon site-specific sedimentation rates and mixing by bioturbation. Therefore, each sample reflects time averages of biomarker production in and out of the polynya. Marine surface sediment samples collected on these cruises as well as other cruises over



the past decades have previously been analyzed for HBIs and select sterols (i.e., brassicasterol and dinosterol) as part of larger
pan-Arctic datasets (Stoynova et al., 2013; Kolling et al., 2020). In addition, Baffin Bay marine surface sediments, including
some of those presented here, have also been analyzed for their mineralogical composition with quantitative X-ray diffractions
(qXRD) in order to track sediment provenance and glacier sources (Andrews and Eberl, 2011; Andrews et al., 2018, Andrews,
2019).

## 4.2. Bulk geochemical proxies

At the University of Colorado Boulder's Earth Systems Stable Isotope Laboratory, freeze dried marine surface sediment
subsamples were analyzed for bulk elemental (%CaCO$_3$, %TC, %TN) and isotopic ($\delta^{13}$C relative and $\delta^{15}$N) geochemistry
following standard procedures on a Thermo Delta V elemental analyzer (EA) interfaced with an isotope ratio mass
spectrometer (IRMS), and computed for elemental C/N values. $\delta^{13}$C values are expressed relative to Vienna Pee Dee Belemnite
(VPDB) and $\delta^{15}$N relative to Air.

## 4.3. Lipid biomarkers

At the University of Colorado Boulder's Organic Geochemistry Laboratory, freeze dried marine surface sediment subsamples
(~1-4 g) were extracted two times on a Dionex accelerated solvent extractor (ASE 200) using dichloromethane
(DCM):methanol (9:1, v/v) at 100 °C and 2,000 psi. A 25 percent aliquot of the total lipid extract (TLE) was taken for GDGT
analysis. The remaining 75 percent of TLE was then separated into five fractions (F1-F5) using silica column chromatography,
after elution with hexane (F1), hexane:DCM (8:2, v/v) (F2), DCM (F3), DCM:ethyl acetate (EtOAc) (1:1, v/v) (F4) and EtOAc
(F5). Each of these extractions contained the following internal standards at 1 ng/μL concentration: 3-methylheneicosane (F1),
p-Terphenyl-d14 (F2), docosanoic acid (F3), 1-nonadecanol (F4), 2-Me octadecanoic acid (F5).

From F1, we focus on highly branched isoprenoids (HBI) IP$_{25}$ (C$_{25:1}$), HBI II (C$_{25:2}$), HBI III (C$_{25:3}$) and HBI IV (C$_{25:3}$)
biomarkers. HBIs were analyzed via gas chromatography-mass spectrometry (GC-MS) on a Thermo Trace 1310 Gas
Chromatograph interfaced to a TSQ Evo 8000 triple quadrupole mass spectrometer and fitted with an Agilent DB-1MS GC
column (60 m x 250 μm x 250 μm) following modified methods and operating conditions of Belt et al. (2012). We used an ion
source temperature of 250 °C rather than 300 °C to prevent excessive fragmentation during ionization, which yields larger
molecular fragment ions that facilitated compound identification both in full (FS) and selected reaction monitoring (SRM)
modes (e.g., Boudinot et al., 2020). The identification and quantification of IP$_{25}$ (*m/z* 350, Belt et al., 2007), HBI II (*m/z* 348,
Belt et al., 2007), and HBI III and IV (*m/z* 346, Belt et al., 2000) was based on their respective mass spectra compared to that
of the internal standard (3-methylheneicosane, *m/z* 310.6) and normalized according to their sediment masses.

From F3, we focus on C$_{37}$ alkenones, which are 37 carbon chain length ketones with either two (C$_{37:2}$), three (C$_{37:3}$)
or four unsaturations (C$_{37:4}$). Alkenones were analyzed via GC-MS fitted with an Agilent VF200-MS GC column (60 m x 250
μm x 0.10 μm) following the operating conditions of Longo et al. (2013). Mass spectrometric analyses were carried out in FS
mode. The identification and quantification of C$_{37:2}$, C$_{37:3}$ and C$_{37:4}$ was based on their respective mass spectra compared to





that of the internal standard (docosanoic acid, *m/z* 340.6) and normalized according to their sediment masses. Using individual alkenone concentrations, we then computed the $U^K_{37}$ and $U^{K'}_{37}$ index after Brassell et al. (1986) and Prahl and Wakeham (1987), respectively:

$$U^K_{37} = \frac{[C_{37:2}]-[C_{37:4}]}{[C_{37:2}]+[C_{37:3}]+[C_{37:4}]},\tag{2}$$

$$U^{K'}_{37} = \frac{[C_{37:2}]}{[C_{37:2}]+[C_{37:3}]},\tag{3}$$

From F4, we focus on a series of sterols, namely brassicasterol (24-Methylcholesta-5,22*E*-dien-3β-ol), dinosterol (4α,23,24-Trimethyl-5α-cholesta-22*E*-en-3β-ol), campesterol (24-Methylcholesta-5-en-3β-ol) and β-sitosterol (24-Ethylcholesta-5-en-3β-ol). Before instrumental analyses, each sample was derivatized with N,O-bis(trimethylsilyl)trifluoroacetamide (BSTFA; 25 μL) and pyridine (catalyst, 25 μL) at 70 °C for 20 min.) Sterols were

analyzed via GC-MS fitted with an Agilent DB-5MS GC column (60 m x 250 μm x 250 μm) under the following operating conditions: initial temperature of 80 °C (held for 2 min.), ramp 20 °C/min. (2.5 min.) , ramp 5 °C/min. (68 min., held for 30 min.). Mass spectrometric analyses were carried out in full scan (FS) and selected reaction monitoring (SRM) modes. Quantification of individual sterols was achieved by comparison of their respective mass spectra (Boon et al., 1979) to that of the internal standard (1-nonadecanol, *m/z* 284.5) and normalized according to their sediment masses.

For GDGTs, we focus on isoprenoid and hydroxylated isoprenoid GDGTs. A 25 percent aliquot of dry TLE samples was resuspended in hexane:isopropanol (99:1, v/v), sonicated, vortexed, and then filtered using a 0.45 μm polytetrafluoroethylene (PTFE) syringe filter. Prior to analysis samples were spiked with 10 ng of the $C_{46}$ GDGT internal standard (Huguet et al., 2006). GDGTs were identified and quantified via high performance liquid chromatography-mass spectrometry (HPLC-MS) following modified methods of Hopmans et al. (2016) on a Thermo Scientific Ultimate 3000 HPLC

interfaced to a Q Exactive Focus Orbitrap-Quadrupole MS (Harning et al., 2019). GDGTs were identified based on their characteristic masses and elution patterns. For isoprenoid GDGTs, we explore the original $TEX_{86}$ index (Schouten et al., 2002) and the more recent $TEX_{86}^L$ index, which is a modification of the former for temperatures <15 °C (Kim et al., 2010, 2012), to reflect relative changes in temperature:

$$TEX_{86} = \frac{[GDGT-2]+[GDGT-3]+[cren.']}{[GDGT-1]+[GDGT-2]+[GDGT-3]+[cren.']},\tag{4}$$

$$TEX_{86}^L = \log\left(\frac{[GDGT-2]}{[GDGT-1]+[GDGT-2]+[GDGT-3]}\right),\tag{5}$$

To evaluate the integrity of GDGT-based temperature estimates, we compute the Ring Index for each sample (RI, Zhang et al., 2016):

$$RI = 0x[GDGT-0] + 1x[GDGT-1] + 2x[GDGT-2] + 3x[GDGT-3] + 4x[cren.] + 4x[cren.'],\tag{6}$$

and compare with the global core top polynomial regression for $TEX_{86}$ values (Zhang et al., 2016):

$$RI_{TEX} = -0.77(\pm0.38)xTEX_{86} + 3.32(\pm0.34)x(TEX_{86})^2 + 1.59(\pm0.10),\tag{7}$$

Finally, for hydroxylated isoprenoid GDGTs, we explore two different relative temperature indices, RI-OH and RI-OH' developed for regions over and under 15 °C, respectively (Lü et al., 2015):



$$RI - OH = \frac{[OH-GDGT-1]+2x[OH-GDGT-2]}{[OH-GDGT-1]+[OH-GDGT-2]}, \qquad (8)$$

$$RI - OH' = \frac{[OH-GDGT-1]+2x[OH-GDGT-2]}{[OH-GDGT-0]+[OH-GDGT-1]+[OH-GDGT-2]}, \qquad (9)$$


### 4.4. WOA18 datasets

To assess and calibrate our lipid biomarkers proxies against modern climatological fields, we use World Ocean Atlas 2018 (WOA18) decadal mean datasets from 2007 to 2017 CE. We compare against various depth integrations (0-200 mbsl) of the following variables: temperature (Locarnini et al., 2018), salinity (Zweng et al., 2018), dissolved oxygen (Garcia et al., 2018a),

and nitrate (Garcia et al., 2018b) (Fig. 2). For each of these variables, we compared proxy data against the mean annual values as well as mean seasonal values where complete datasets were available. Since winter and spring seasonal data were either fragmentary or not available, the mean annual datasets used in this study likely represent the ice-free season, integrating summer and shoulder season months.

### 5. Results

### 5.1. Bulk geochemical proxies

Although only nine out of the 13 total surface sediment samples were analyzed for bulk geochemistry, those that were represent the full geographical range of sample types throughout Baffin Bay (Fig. 1). Bi-plots of $\delta^{13}C$ and C/N tightly cluster within the range of marine algae (Fig. 3, Meyers, 1994).

### 5.2. HBIs

HBIs were present above the detection limit in all sediment samples. $IP_{25}$ was the most dominant HBI in the dataset, followed by HBI II, HBI III, and HBI IV (Fig. 4a). HBIs were generally more abundant at NOW sites compared to sites outside this region, although there was some overlap when the standard deviation of mean concentrations was considered for $IP_{25}$, HBI II and HBI IV, but not for HBI III (Figs. 4a and S1). Based on Pearson correlation analysis, each of the HBIs strongly correlated

with the others (>0.74, Fig. 5).

### 5.3. Sterols

Sterols were present above the detection limit in all sediment samples. ß-sitosterol was the most abundant sterol in the dataset, followed by campesterol, brassicasterol, and dinosterol (Fig. 4b). Sterols were generally more abundant at NOW sites

compared to sites outside this region (Fig. 4b and S2). Although the standard deviations of mean dinosterol and campesterol concentrations overlapped between NOW and non-NOW sites, the standard deviations of ß-sitosterol and brassicasterol for the two regions were statistically different (Fig. 4b and S2). Based on Pearson correlation analysis, the sterols all strongly correlated with each other (>0.72), apart from dinosterol and ß-sitosterol whose correlation was insignificant (Fig. 5).





### 5.4. PIP$_{25}$


For the PIP$_{25}$ indices, we calculated new Baffin Bay balance factors (c) for P$_{III}$IP$_{25}$ (2.11), P$_{IV}$IP$_{25}$ (2.49), P$_B$IP$_{25}$ (0.52), and P$_D$IP$_{25}$ (0.73) based on a combination of 139 surface sediment samples from the region, including Hudson Strait and the Gulf of St. Lawrence (Kolling et al., 2020), and our samples that reflect conditions in the NOW. Although HBI IV was quantified by Kolling et al. (2020), we note that the authors did not explore its potential as part of a PIP$_{25}$ index. For all four PIP$_{25}$ indices,

we observed slightly higher mean values in non-NOW sites compared to NOW sites, although the standard deviation of mean values between these regions were not statistically different (Fig. 4c and S3).

### 5.5. Alkenones

Alkenones were present above the detection limit in 9 out of 13 sediment samples. C$_{37:2}$ was the dominant alkenone, followed

by C$_{37:4}$ and then C$_{37:3}$ (Fig. 4d). Although relative abundances of alkenones are similar between sites, alkenone concentrations were higher at NOW sites compared to those farther south in Baffin Bay (Fig. 4d).

In terms of regressions against different environmental variables (e.g., temperature, salinity, dissolved oxygen, and nitrate), we find that the U$^K_{37}$ and U$^{K'}_{37}$ indices ranged in the strength of correlations. Moreover, differences in the season and depth integration of the environmental variables also appeared to influence the strength of index correlations (Fig. 6). For U$^K_{37}$,

the strongest temperature correlations were achieved with summer SST ($R^2$ = 0.39 to 0.61, Fig. 6a), while no correlations existed with annual SST ($R^2$ < 0.02, Fig. 6a). Depending on the season, salinity and nitrate concentrations featured low to high correlations with U$^K_{37}$ ($R^2$ = 0.07 to 0.71, salinity; $R^2$ = 0.12 to 0.87, nitrate) (Figs. 6c and 6e). For U$^{K'}_{37}$, the strongest temperature correlation was achieved with annual SST ($R^2$ = 0.37 to 0.58, Fig. 6b). The influence of annual and summer salinity on U$^{K'}_{37}$ appeared to be weak ($R^2$ < 0.14 Fig. 6d), whereas moderate correlations were found with annual nitrate above 10 m

bsl ($R^2$ = 0.44 to 0.56, Fig. 6f). Finally, we found weak to strong correlations between the fractional abundance of C$_{37:4}$ alkenones and salinity, particularly during the summer and at the upper surface ($R^2$ = 0.16 to 0.73, annual; $R^2$ = 0.74 to 0.92, summer, Fig. 6c inset).

For our strongest annual U$^{K'}_{37}$ temperature correlation that integrates the upper 20 m of the water column (Fig. 6b), we derived a linear temperature calibration for northern Baffin Bay. This local U$^{K'}_{37}$ calibration features a moderate $R^2$ of 0.58

and a low standard error (S.E.) of 0.30 °C (Fig. 7a):

$$SST = 12.216x - 9.7355, \tag{10}$$

For reasons discussed in later Section 6.2, we did not generate a temperature calibration based on the U$^K_{37}$ index.

### 5.6. GDGTs and OH-GDGTs

GDGTs and OH-GDGTs were present above the detection limit in all sediment samples. In terms of RI values, all samples plotted below the global core top polynomial regression for TEX$_{86}$ (Fig. 8a). In terms of total fractional abundance, GDGTs (0.88 ± 0.03) dominated over their OH-GDGT counterparts (0.12 ± 0.03) (Fig. 8b). GDGTs and OH-GDGTs with no





cyclopentane moieties were the most dominant (e.g., GDGT-0 and OH-GDGT-0), followed by crenarchaeol and then the GDGTs and OH-GDGTs with 1, 2, and 3 cyclopentane moieties in the case of GDGTs (Fig. 8c-d).

295       In terms of regressions against different environmental variables (e.g., temperature, salinity, dissolved oxygen, and nitrate), we found that the tested GDGT ($TEX_{86}$ and $TEX_{86}^L$) and OH-GDGT-based indices (RI-OH and RI-OH') ranged in the strength of the correlation coefficients. Moreover, differences in the season and depth integration of the environmental variables also appeared to influence the strength of index correlations (Figs. 9, S4, S5, and S6). For $TEX_{86}$, the strongest temperature correlation was achieved with summer SST above 20 m bsl ($R^2$ = 0.21 to 0.35, Fig. 9a). Although salinity and

dissolved oxygen seem to have little confounding influence ($R^2 < 0.22$), we found a moderate correlation between annual $TEX_{86}$ and nitrate concentrations above 5 m bsl ($R^2$ = 0.40, Fig. S6), similar to temperature correlations. For $TEX_{86}^L$, moderate temperature correlations ($R^2 > 0.40$) were found with autumn (>80 m bsl) and annual seasons (>30 m bsl), whereas the other shallower depths, as well as spring, featured weak correlations (Fig. 9b). The influence of other environmental variables (e.g., salinity, dissolved oxygen, and nitrate) on $TEX_{86}^L$ appeared to be generally weak ($R^2 < 0.27$), although summer nitrate

concentration correlations at 0-200 m bsl depth were slightly higher ($R^2$ = 0.37, Fig. S6). For RI-OH, the strongest temperature correlations were achieved with annual SST between 10 and 20 m bsl ($R^2$ = 0.44 to 0.46), although autumn subsurface temperature between 60 and 80 m bsl also featured similar moderate correlations ($R^2$ = 0.41 to 0.42) (Fig. 9c). The influence of the other tested environmental variables on RI-OH appeared to be generally weak ($R^2 < 0.27$), although summer and annual 0-20 m dissolved oxygen concentration correlations were slightly higher ($R^2$ = 0.39 and $R^2$ = 0.36, respectively, Fig. S5).

Finally, for RI-OH', the strongest, albeit weak, temperature correlations were achieved with annual 0-20, annual 0-30 m SST, and autumn 0-100 m SST ($R^2$ = 0.23) (Fig. 9d). Based on our $R^2$ values, the influence of other environmental variables, namely annual and summer subsurface dissolved oxygen and nitrate below 100 m depth, appeared to exert relatively more influence than temperature on RI-OH' (Figs. S5 and S6).

      For the two indices that featured moderate correlations with temperature ($R^2 > 0.40$), we derived individual linear

temperature calibrations for northern Baffin Bay. The $TEX_{86}^L$ calibration ($R^2$ = 0.45) encompasses the surface and subsurface waters from 0-90 m bsl and features a standard error (S.E.) of 0.13 ℃ (Fig. 10a):

$$subT = 10.573x + 6.636, \tag{11}$$

The RI-OH calibration ($R^2$ = 0.46) encompasses the surface waters from 0-20 m bsl and features a S.E. of 0.26 ℃ (Fig. 10b):

$$SST = 27.982x - 31.524, \tag{12}$$

## 320   6. Discussion

### 6.1. Spatial variability in surface productivity

A recent and expanded Arctic study by Kolling et al. (2020) explored the efficacy of using a variety of sedimentary HBIs (i.e., $IP_{25}$, HBI II, and HBI III) and sterols (i.e., brassicasterol and dinosterol) and their $PIP_{25}$-derived indices to reconstruct sea ice extent/concentration and pelagic productivity by comparison to satellite data. For Baffin Bay, the main conclusions were that

$P_BIP_{25}$, $P_DIP_{25}$ and $P_{III}IP_{25}$ all exhibited strong correlations with modern spring and autumn seasonal sea ice concentration



(Kolling et al., 2020). On the other hand, TR$_{25}$, the recently proposed HBI proxy for spring phytoplankton bloom in the Barents Sea (Belt et al., 2019), did not exhibit a clear relationship with chlorophyll *a*, but instead paralleled spring/autumn/winter sea ice extent (Kolling et al., 2020). For this reason, we did not further explore the controls of chlorophyll *a*, which is a limited and discontinuous dataset in the Baffin Bay region (NASA MODIS), on the distribution of HBI III and HBI IV compounds

that comprise the TR$_{25}$ proxy (e.g., Belt et al., 2019). However, we do further explore the relationship of HBIs, sterols and PIP$_{25}$ indices in their ability to track sea ice extent, and more importantly, the elevated surface productivity associated with the NOW. Moreover, we include several additional sterols (ß-sitosterol and campesterol) and introduce another PIP$_{25}$ index (P$_{IV}$IP$_{25}$) that have not previously been tested in Baffin Bay.

In terms of HBI concentrations, all compounds were considerably higher in sites within the NOW compared to sites

farther south, and outside the limit of the NOW (Fig. 4a and S1). For IP$_{25}$ and HBI II, the detection of each in all samples is consistent with the spring and autumn sea ice extents in Baffin Bay (Fig. S1, Bi et al., 2019). Unlike Kolling et al. (2020), but similar to Amiraux et al. (2021), we observe a strong positive correlation between IP$_{25}$ and HBI II (Fig. 5), which supports a common sympagic diatom source as observed elsewhere in the Arctic (Belt et al., 2008; Cabedo-Sanz et al., 2013; Brown et al., 2014; Limoges et al., 2018). Given the distinct oceanographic seasonal characteristics of the NOW (open water) and central

Baffin Bay (seasonal sea ice), we would expect higher IP$_{25}$ and HBI II concentration in the sites south of the NOW compared to sites within the NOW itself. Although seemingly counterintuitive, the higher concentrations of IP$_{25}$ and HBI II within the NOW may simply reflect the transport of drift ice through the NOW from Smith Sound to the Baffin Bay Current, and the generation and deposition of IP$_{25}$ and HBI II by the sympagic diatom that the sea ice supports en route. Alternatively, this could also reflect the lateral transport of HBIs from around the NOW and/or the higher export of marine organic matter in the

NOW compared to less productive areas in Baffin Bay. Although in relatively lower concentrations, the distribution of HBI III and IV is similar to IP$_{25}$ and HBI II (Fig. S1), which presents several possibilities. First, as observed in other regions of the Arctic such as the Barents Sea, increased concentrations of HBI III and IV are associated with the open water of Marginal Ice Zones (Smik et al., 2016; Belt et al., 2015, 2019). In this sense, the concentrations we observe in Baffin Bay would be consistent with increased production of these HBIs during the spring transition from seasonal sea ice to open water in the NOW compared

to sites outside that maintain seasonal sea ice cover during the summer. Alternatively, recent work that has monitored the production of various HBIs through the spring sea ice melt season in southwest Baffin Bay shows that HBI III is produced under the sea ice before and concurrent with IP$_{25}$, and therefore, is likely biosynthesized by some sympagic diatoms in this region (Amiraux et al., 2019, 2021). Given that all HBIs show no significant correlations with sterols, which reflect open water productivity, we argue that the latter scenario where HBI III and IV are produced by sympagic diatoms is most likely (Fig. 5).

In terms of sterol and alkenone concentrations, all compounds were considerably higher in sites within the NOW compared to sites farther south, and outside the limit of the NOW (Fig. 4b and 4d). While a recent Holocene marine record from Petermann Fjord (Northwest Greenland) interprets campesterol and ß-sitosterol as indicators of terrestrial input (Detlef et al., 2021), we note that in our datasets all sterols are highly correlated (Fig. 5) suggesting a common marine source, consistent with corresponding bulk geochemistry data (Fig. 3). Although the sterols do not feature the same degree of source specificity



that the HBIs do, the spatial variability of our results is broadly consistent with the high seasonal biological productivity that

the NOW supports. Moreover, studies based on nutrient concentrations of particulate matter in Baffin Bay show that primary

productivity in the NOW is an order of magnitude higher than adjacent areas in Baffin Bay during spring/summer (Tremblay

et al., 2002a). The localized and high concentrations of sterols in the NOW compared to non-NOW sites is broadly consistent

with these trends (Fig. 4b and S2). Given the propagation of sea ice southward, and that the development of the NOW occurs

at the transition between spring and summer (Bi et al., 2019), the sterols are likely recording a spring/summer season signal of

pelagic productivity. The one site within the NOW that does not exhibit high sterol concentrations is at the northern limit, and

therefore either may not receive the full effects of nutrient delivery from the WGC and/or is still under the influence of residual

sea ice drift or Smith Sound blockage during the summer months. In a similar sense, we also observe generally higher

concentrations of sterols in the central NOW compared to sites on the NOW periphery abutting Jones and Lancaster Sounds

that may feature more persistent seasonal sea ice presence than those in the center (Fig. S2).

Finally, in terms of $PIP_{25}$, all four indices ($P_{III}IP_{25}$, $P_{IV}IP_{25}$, $P_BIP_{25}$, and $P_DIP_{25}$) were lower in sites within the NOW

compared to sites farther south, and outside the limit of the NOW (Fig. 4c and S3). While the HBI-derived indices ($P_{III}IP_{25}$ and

$P_{IV}IP_{25}$) feature higher values for both the NOW and non-NOW sites compared to those derived from sterols ($P_BIP_{25}$, and

$P_DIP_{25}$), we caution against the application of the HBI-derived indices in Baffin Bay for sea ice concentration since these

biomarkers may all originate from sea ice diatoms (Amiraux et al., 2019, 2021), rather than a combination of sea ice and open

water diatoms. Looking at the sterol-derived indices ($P_BIP_{25}$, and $P_DIP_{25}$), their values with respect to qualitative sea ice

concentrations defined by Müller et al. (2011) for sites in the northern North Atlantic indicate that the NOW sites feature

seasonal sea ice cover (0.5 to 0.7) whereas non-NOW sites feature seasonal ($P_DIP_{25}$, 0.5 to 0.7) to perennial ($P_BIP_{25}$, 0.7 to 1)

sea ice cover (Fig. 4c). This spatial variability is consistent with the spring/late autumn sea ice cover in Baffin Bay, with

highest $PIP_{25}$ index values off eastern and northeastern Baffin Island where sea ice persists the longest during the spring and

summer (Bi et al., 2019). Moreover, these analyses help verify the capability of $P_BIP_{25}$ and $P_DIP_{25}$ in tracking seasonal

(spring/late autumn) sea ice cover (e.g., Kolling et al., 2020) and highlight their strength as paleoenvironmental proxies in

Baffin Bay.

In summary, we have several recommendations for future paleoenvironmental reconstructions in Arctic

oceanographic settings. First, for complex oceanographic settings like Baffin Bay, we recommend the analysis of both sterols

and HBIs to test the performance of various sea ice cover proxies. If possible, this would be best achieved through the analysis

of modern surface sediments and sediment traps over an environmental gradient to capture proxy response to known variable

changes (e.g., Navarro et al., 2013; Smik and Belt, 2017; Koch et al., 2020). Second, continued research on sterol and HBI

sources and seasonality of production is critical for the development of more refined sea ice and marine productivity

reconstructions (e.g., Limoges et al., 2018; Amiraux et al., 2019, 2021). This is particularly necessary when combining $IP_{25}$, a

known sea ice diatom proxy, with presumably more complicated open water proxies such as HBI III and sterols. Based on our

dataset, which shows an order of magnitude higher sterol concentrations within the NOW relative to sites outside of its modern



limits, compared to HBIs that show only slighter higher concentrations within the NOW, we suggest that sterols, rather than HBIs, may be a more appropriate tool to characterize the presence/absence of the NOW in the recent geologic past.


## 6.2. Temperature calibrations

Temperature correlations for alkenones and GDGTs first relied on empirical correlations between global surface sediments and the variability in biomarker structure (e.g., Brassell et al., 1986; Schouten et al., 2002). Subsequent iterations and developments of $U^K_{37}$ and $TEX_{86}$-based indices removed some compounds (e.g., $C_{37:4}$ for $U^{K'}_{37}$ and crenarchaeol regioisomer

for $TEX_{86}^L$) for improved performance in certain latitudinal bands (e.g., Prahl and Wakeham, 1987; Kim et al., 2012). Most recently, Bayesian statistics have been employed to generate spatially varying calibrations (e.g., BAYSPAR and BAYSPLINE) that reach a compromise between data-constrained global and regional calibrations in order to more accurately capture regional oceanographic variability and site-specific uncertainty (Tierney and Tingley, 2014, 2018). However, the latter calibrations (BAYSPAR and BAYSPLINE) do not include surface sediment for sensitive oceanographic regions like Baffin Bay (Tierney

and Tingley, 2014, 2018). Moreover, given the high $U^{K'}_{37}$-temperature residuals for locations affected by sea ice cover, BAYSPLINE is not advised for many high-latitude regions (Tierney and Tingley, 2018). For this reason, developing local correlation-constrained calibrations can be particularly useful to capture the nuances of oceanographic variability in regions where global dataset coverage is lacking or where the temperature relationship deviates from global linear calibrations (e.g., Harning et al., 2019; Park et al., 2019; Fietz et al., 2020). Moreover, exploring the potential influence of additional

environmental variables (e.g., salinity, DO, nutrients) will help us better understand the mechanisms behind and functionality of these commonly applied "paleotemperature" proxies. Given the relatively small sample size of our dataset compared to global compilations (e.g., Müller et al., 1998; Kim et al., 2012; Lü et al., 2015), we acknowledge that our conclusions in this study are limited. However, as additional data is added from Baffin Bay and other high-latitude regions, we will be better poised to test these observations.


### 6.2.1. Alkenones

In general, our Baffin Bay $U^K_{37}$ and $U^{K'}_{37}$ values are high compared to other similar latitude locations in the greater North Atlantic region (e.g., Rosell-Melé et al., 1994; Müller et al., 1998, Bendle and Rosell-Melé, 2007; Moros et al., 2016; Bartels et al., 2017; Kristjánsdóttir et al., 2017; Łącka et al., 2019). Given that alkenone distributions can also be affected by nutrients,

light limitation, salinity and algae source, it is possible that our high $U^K_{37}$ and $U^{K'}_{37}$ index values relate to some combination of these environmental factors. To further explore this possibility, we compared Baffin Bay $U^K_{37}$ and $U^{K'}_{37}$ index values against temperature as well as salinity and nitrate values from each of our sample locations. Based on our datasets, summer salinity exhibited the strongest correlation with $U^K_{37}$ (Fig. 6a), and sea surface nitrate concentrations exhibited the strongest correlations with $U^{K'}_{37}$ values (Fig. 6f). While $U^{K'}_{37}$ also had comparably moderate correlations with annual SST, as it did with

nitrate concentrations, $U^K_{37}$ values correlated more strongly with salinity than SST (Fig. 6). The latter is likely controlled by the inclusion of $C_{37:4}$, whose fractional abundance features a strong relationship with annual ($R^2 = 0.73$) and summer salinity





($R^2 = 0.92$) in Baffin Bay (Fig. 6c inset). This notion is supported by a lack of correlation between $U^{K'}_{37}$, which excludes $C_{37:4}$ (Prahl and Wakeham, 1987), and salinity (Fig. 6d). Collectively, this evidence indicates that temperature, salinity, and nitrate are all compounding factors that influence $U^{K}_{37}$ and $U^{K'}_{37}$ values in in Baffin Bay, which may complicate a straightforward

extraction of temperature estimates. However, given our limited dataset at this time, temperature seems to be the most important environment variable on $U^{K'}_{37}$ values (Fig. 6).

       The general relationship that $C_{37:4}$ abundance increases with lower salinities has previously been observed in surface sediments in the Nordic (Rosell-Melé, 1998; Sicre et al., 2002) and Bering (Harada et al., 2003) Seas, and has been applied as a qualitative paleo-salinity proxy in certain oceanic regions (Seki et al., 2005), including off west Greenland (Moros et al.,

2016). However, recent biomarker and genetic evidence from the Canadian Arctic Archipelago (CAA) suggests that much of this variability may be related to a previously unrecognized alkenone producer, the sea-ice dwelling Group 2i Isochrysidales lineage, which suggests that $C_{37:4}$ abundance may instead serve as a sea ice proxy (Wang et al., 2021). While we lack genetic information to pinpoint the biological producers of alkenones in Baffin Bay, a comparison between $IP_{25}$ and $C_{37:4}$ abundance in our dataset reveals no correlation (Fig. S7), suggesting that sympagic haptophytes may not be contributing significant

amounts of $C_{37:4}$ to the local alkenone pool. This notion is further supported by the distinct relative abundances of $C_{37:4}$ produced by CAA Group 2i Isochrysidales (0.59 to 0.68) compared to our Baffin Bay surface sediment samples (0.15 to 0.44), which also span the range observed in the marine sediment record from west Greenland (Moros et al., 2016). However, we do note that $C_{37:4}$ values in Baffin Bay are generally higher than those in non-polar regions, which may possibly suggest a mixture of alkenones derived from pelagic and sympagic algal producers. In any case, future efforts aimed at disentangling different

alkenone sources are needed and would benefit from a combined biomarker-DNA analysis such as Wang et al. (2021).

       After acknowledging these strengths and limitations, we propose one $U^{K'}_{37}$-SST calibration that may prove useful for future paleoceanographic research in the region (Fig. 7a). Given that $U^{K'}_{37}$ excludes the $C_{37:4}$ alkenone, it offers some advantage for temperature reconstructions, as it minimizes the confounding influence of salinity and/or sea ice haptophyte source on alkenone distributions. Although the $R^2$ of 0.58 for our Baffin Bay $U^{K'}_{37}$-SST correlation is considerably lower than the global

calibration ($R^2 > 0.95$, Müller et al., 1998), this may simply result from our relatively small sample set size ($n = 9$), or mixed contributions yet unknown haptophyte sources. We posit that the similar degree of correlation observed between $U^{K'}_{37}$ and nitrate and between $U^{K'}_{37}$ and temperature is not necessarily a confounding variable, but more likely represents that preferred habitat of the alkenone-producing haptophytes at the surface where sunlight needed for photosynthesis is available. Although alkenone production is prohibited during winter months due to the lack of sunlight, the better performance of our annual

calibration over summer likely reflects the increased productivity during the summer shoulder seasons of spring and autumn observed in Baffin Bay today (e.g., Tremblay et al., 2002a).

       Due to the relatively high $U^{K'}_{37}$ values for the latitude, annual SST estimated with the global calibration (Müller et al., 1998) result in considerably higher temperatures (Fig. 7b), with residuals from WOA18 0-20 m SST ranging from ~21 to 25 ºC (Fig. 7c). However, when estimates are calculated using our local annual SST calibration, values are in line with the

annual WOA18 data (Fig. 7b), and residuals are reduced to <1.07 ºC (Fig. 7c). Moreover, the S.E. of the calibration is reduced





from 1.1 °C in the global dataset (Müller et al., 1998) to 0.30 °C in our Baffin Bay SST dataset (Fig. 7a). Both the reduced residuals and S.E. highlight one strength in generating local temperature calibrations for high latitude and oceanographically complex locations like Baffin Bay, and presents an opportunity to apply this calibration in the paleo record.

**6.2.2. GDGTs and OH-GDGTs**

While a more detailed analysis of intact polar lipid production and genetic diversity in Baffin Bay is lacking, the distribution of GDGT and OH-GDGT in our study area (Fig. 8) and understanding of their production in cultures (Elling et al., 2017) indicate that planktic group 1.1a Thaumarchaeota are likely the dominant producers. Therefore, the global relationship between $TEX_{86}$ and the RI (Fig. 8a, black polynomial line) serves as a simple means to evaluate whether the $TEX_{86}$-based indices are

influenced by additional nonthermal factors (Zhang et al., 2016). Even though our data exhibit a correlation between $TEX_{86}$ and RI values ($R^2 = 0.46$) with a slope like the global polynomial regression, all samples plot below the lower 95% uncertainty limit (Fig. 8a). A recent study from the South China Sea found a similar relationship to ours, in that $TEX_{86}$ values were well correlated with RI values, but did not conform to the global polynomial trend's uncertainty (Wei et al., 2020). Wei et al. (2020) posited that the shallow shelf environment (neritic zone) of the South China Sea may result in the observed deviation from the

global polynomial $TEX_{86}$-RI relationship, as shallow water Thaumarchaeota respond differently to temperature than the deep-dwelling communities (e.g., Kim et al., 2015, 2016; Villanueva et al., 2015; Zhu et al., 2016; Jia et al., 2017). While the depths of our sites are deeper than the neritic zone (>200 m bsl, Table 1), many are much shallower than open ocean sites used in the global calibrations (Kim et al., 2010, 2012). One possibility is that the bay-like environment of northern Baffin Bay may result in a different response of Thaumarchaeota to temperature and different distribution of GDGTs than in the open ocean. In any

case, temperature seems to remain the dominant control on GDGT cyclization in this region.

Our regression analysis against temperature, salinity, DO and nitrate further supports temperature as the dominant environmental control on GDGT distributions in Baffin Bay for the seasons available in the WOA18 dataset. The lack of WOA18 winter temperatures, in addition to the fragmentary dataset for spring temperatures in Baffin Bay (Locarnini et al., 2018), prevents us from assessing the impact of these individual seasons, which is unfortunate given that cold season

temperatures in other high-latitude settings exhibit a stronger correlation with GDGT distributions (Herfort et al., 2006; Rueda et al., 2009; Rodrigo-Gámiz et al., 2015; Harning et al., 2019). However, the higher correlation between $TEX_{86}^L$ and annual temperatures compared to summer/autumn seasons (Fig. 9b) suggests a bias for Baffin Bay GDGT production towards the cooler shoulder seasons. Although we explored the relationship of the original $TEX_{86}$ index, we prefer to rely on the low-temperature $TEX_{86}^L$ modification to assess GDGT depth and season and temperature relationships. This decision is supported

by 1) generally low correlation coefficients for temperature and $TEX_{86}$, except for summer SST (Fig. 9a), and 2) the better correlations of $TEX_{86}^L$ and lower water depths where ammonia-oxidation likely occurs (Fig. 8b, e.g., Hurley et al., 2018, Park et al., 2019). The moderate correlation between $TEX_{86}$ and summer SST, and the lack of correlation between $TEX_{86}^L$ and SST from any season, may indicate that the crenarchaeol regioisomer, which is included in the $TEX_{86}$ index but not in $TEX_{86}^L$, is





physiologically advantageous for surface dwelling Thaumarchaeota during the warmer summer months in Baffin Bay. We can
also not rule out the possibility of a yet unknown alternative biological source.

         In terms of other tested environmental variables, correlations between $TEX_{86}^L$ and salinity and dissolved oxygen are
generally poor ($R^2 < 0.3$, Figs. S4 and S5), consistent with previous reports on the lack of relationship between GDGT
production and salinity (Wuchter et al., 2004, 2005; Elling et al., 2015). The presence of a well-oxygenated water column in
Baffin Bay (Fig. 2c) likely has little influence on GDGT cyclization, which has been reported to only occur in oxygen-limited
environments (e.g., Qin et al., 2015). The correlation between $TEX_{86}^L$ and nitrate, while weak, is more pronounced in the
lowermost integrated depth (200 m bsl) (Fig. S6). The main product of ammonia-oxidation is nitrate, for which we have no
data for in this region. However, since nitrate can be subsequently oxidized to nitrate by bacteria (Kuypers et al., 2018), we
assume that nitrate provides indirect evidence for both reactions. This allows us to confirm the subsurface depth habitat of
ammonia oxidizing Thaumarchaeota in Baffin Bay, which supports our observations that $TEX_{86}^L$ correlates best with annual
subT (Fig. 9b).

         Our complimentary analysis of OH-GDGTs and environmental variables reveals several differences with the
conclusions drawn about GDGTs in Baffin Bay and with OH-GDGTs elsewhere. First, the RI-OH' index, which was developed
as a modified version of RI-OH for low-temperature environments (Lü et al., 2015), does not perform as well as the RI-OH
index in our dataset (Fig. 9c-d). Second, we find that RI-OH is best correlated with annual SST ($R^2 = 0.46$), although during
autumn the subsurface also appears to be important (Fig. 9c). The stronger correlation between RI-OH and SST rather than
subT that we observe is supported by Lü et al. (2019), who observed higher concentrations of OH-GDGTs in the upper portion
of the water column compared to GDGTs in the East China Sea. Additional studies have found that the ring composition of
OH-GDGT often differs from GDGTs, which may suggest that these two lipid classes are sourced from different
Thaumarchaeota subgroups or produced under different environmental conditions (Liu et al., 2012).

In terms of additional environmental variables, only surface dissolved oxygen appears to exert a partial influence on
OH-GDGT cyclization in the RI-OH index (Fig. S5c). However, when OH-GDGT-0 is considered in the RI-OH' index,
moderate correlations with dissolved oxygen (Fig. S5d) and nitrate appear within the subsurface waters (Fig. S6d). These latter
observations conflict with our RI-OH temperature correlations that suggest a surface-dwelling producer of OH-GDGTs.
Instead, this may suggest that OH-GDGT-0 is important for membrane functionality at these depths and/or for these
environmental variables. Alternatively, OH-GDGT-0 may also have a different biological source in Baffin Bay, as at least for
GDGTs, GDGT-0 can be produced by other groups of archaea (e.g., Pancost et al., 2001). Given the relatively low correlations
between RI-OH', which includes OH-GDGT-0, and temperature, our data further supports the potential role of OH-GDGT-0
for membrane response to these non-thermal environmental variables in Baffin Bay. Although our data suggests that OH-
GDGTs, reflected by the RI-OH index, are predominately a proxy for annual SST, more detailed studies on the biological
producers, depth habitat, and response to environmental variables of OH-GDGTs will undoubtedly benefit future applications
and interpretations of downcore OH-GDGT proxy records.





Following our evaluation of GDGT and OH-GDGT indices in terms of their ability to capture temperature, amongst other environmental variables, we present two temperature calibrations that may benefit future paleoceanographic reconstructions from Baffin Bay (Fig. 9). Similar to other regional $TEX_{86}^L$ temperature calibrations in the greater North
Atlantic region (e.g., Harning et al., 2019), our Baffin Bay $TEX_{86}^L$ temperature calibration captures a subsurface signal (90 m bsl) with a considerably lower S.E. (0.13 °C) compared to the latest global calibration (4.0 °C, Kim et al., 2012). The strength of our local $TEX_{86}^L$ subT calibration is also supported by the fact that other regional (Harning et al., 2019) and global calibrations (Kim et al., 2012) overestimate the observed WOA18 temperature in Baffin Bay (Fig. 10c), with residuals as high as 5.8 °C for the Iceland calibration and as high as 5.0 °C for the global calibration (Fig. 10e). In terms of our RI-OH SST
calibration, to the best of our knowledge there are no regional calibrations to compare with. However, comparison with the global calibration that features a S.E. of 6.0 °C (Lü et al., 2015) again reveals the power of reducing uncertainty with regional calibrations as ours from Baffin Bay features a smaller S.E. of 0.26 °C. While the global $TEX_{86}^L$ calibration overestimated Baffin Bay temperatures by up to 5.0 °C, the RI-OH calibration only overestimates local temperatures by less than 3.3 °C (Fig. 10d-f). In future studies, we suggest that these two calibrations may bolster the quantification of paleotemperature change in
Baffin Bay surface and subsurface waters. For other high-latitude locations that do not yet have local GDGT-temperature calibrations, our comparison between instrumental temperature and global calibration estimates caution against quantitative interpretations due to large over estimations. However, as the trends between regional and global calibrations are similar (Fig. 10c-d), qualitative interpretations remain valid.

## 7. Conclusions

As global climate change continues in the current century, the sustainability of Arctic polynyas is in jeopardy. While proxy reconstructions of polynyas prior to the instrumental period can shed light on key climate and environmental mechanisms that lead to their presence/absence, a detailed understanding of those proxies is needed. Therefore, we evaluated a series of lipid biomarkers (HBIs, sterols, alkenones, GDGTs, OH-GDGTs) in surface sediment samples from Baffin Bay to characterize how these biomarkers capture sea ice and productivity conditions in the North Water Polynya (NOW) as well as inform the utility
of commonly applied paleotemperature proxies in Baffin Bay.

- All HBIs ($IP_{25}$, HBI II, HBI III, and HBI IV) exhibit strong correlations with each other and are found in highest concentrations within the modern limits of the NOW. We suggest all HBIs are, at least partially, produced by sympagic diatoms under the sea ice that is formed in the NOW during spring and autumn, and subsequently advected southward along the coast of eastern Baffin Island.

- All studied sterols (dinosterol, campesterol, ß-sitosterol, and brassicasterol) exhibit an order of magnitude higher concentrations within the NOW compared to sites further south in Baffin Bay, consistent with the order of magnitude higher spring/summer primary productivity that is observed within the NOW today relative to surrounding waters. Hence, we suggest that sterols, rather than HBIs, are more suitable for open water paleoproductivity reconstructions in this region.





•   All alkenones ($C_{37:2}$, $C_{37:3}$, and $C_{37:4}$) exhibit highest concentrations within the NOW, which likely reflects the higher productivity of algal haptophytes during spring/summer blooms. Due to high relative abundances of $C_{37:4}$, which may be influenced by salinity and/or sea ice haptophyte sources, we suggest that the $U^{K'}_{37}$ temperature proxy is more appropriate for application in Baffin Bay. We present a local annual SST calibration that provides a reduced S.E. (0.30 °C) compared to global datasets (1.1 °C, Müller et al., 1998) and may benefit future paleoceanographic

reconstructions in the region.

    •   Application of the GDGT-based $TEX_{86}^{L}$ index is optimal over $TEX_{86}$ in Baffin Bay, and captures an integration of annual subT consistent with the depth habitat of Thaumarchaeota observed elsewhere around the globe. Other tested environmental variables, such as salinity, dissolved oxygen, and nitrate reveal low correlations, which we infer to reflect negligible influence on GDGT cyclization. We present a local annual subT calibration that provides a

considerably lower S.E. (0.13 °C) compared to the latest global calibration (4.0 °C, Kim et al., 2012).

    •   Application of the OH-GDGT-based RI-OH index is optimal over RI-OH', even though the latter was developed for low temperature environments like Baffin Bay. The RI-OH index captures an integration of annual SST, which may suggest that OH-GDGTs in Baffin Bay are biosynthesized by different microbes than those that produce GDGTs or are sensitive to different environmental stresses. Our local annual SST calibration provides a considerably lower S.E.

(0.26 °C) compared to the latest global calibration (6.0 °C, Lü et al., 2015).

## Data Availability

After acceptance, all data will be stored on the PANGAEA repository, and available upon reasonable request to the authors.

## Author Contributions

JS and DJH designed the study. AEJ obtained sediment samples. JS and AEJ funded the study. DJH led the analyses of samples and developed GC-MS methods under the supervision of JS. BH and LW assisted with the extraction and purification of samples. DJH wrote the manuscript with discussion and contribution from all co-authors.

## Competing Interests

The authors declare they have no conflicts of interest.

## Acknowledgements


We kindly thank the captains, crews, and scientific staffs aboard the 2008 CSS *Hudson* and 2017 CSS *Amundsen* research cruises for their efforts in collecting the surface sediment samples. We appreciate the valuable analytical support of Dr. Nadia





Dildar at the University of Colorado Boulder. This project has been supported by the National Science Foundation grant ARN-1804504.

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

**Figure 1: Overview map of Baffin Bay. Simplified ocean surface currents shown in bold lines where red reflects warm, Atlantic Water (West Greenland Current, WGC) and blue reflects cool, Arctic Water (Baffin Current, BC). To the north is the June limit of**





the NOW (purple dotted line). Seasonal sea ice limits shown with black dotted (autumn), dashed (spring) and solid lines (winter) (Cavalieri et al., 1996). The locations of modern marine surface sediments are shown with red circles. CAA = Canadian Arctic Archipelago, NS = Nares Strait, SS = Smith Sound, JS = Jones Sound, and LS = Lancaster Sound. See Table 1 for further sample site information.

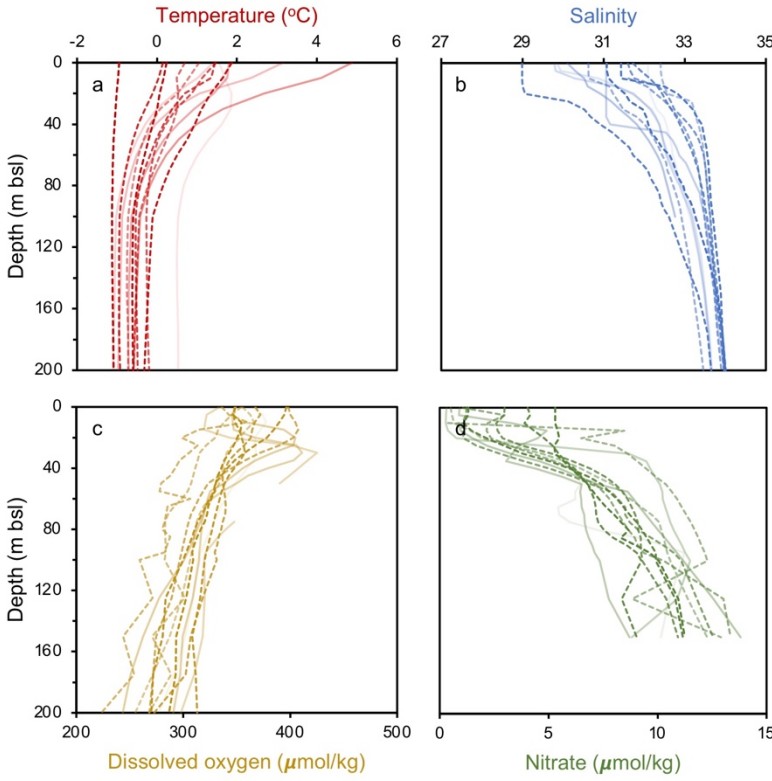

**Figure 2: WOA18 Annual 2007-2017 oceanographic variables from Baffin Bay against depth (m bsl). Individual profiles are from**
**each of our sites, where darker (lighter) colors reflect sites farther north (south) and dashed (solid) lines denote those within (outside)**
**the modern limits of the NOW. Data from Garcia et al. (2018a, 2018b), Locarnini et al. (2018), and Zweng et al. (2018).**




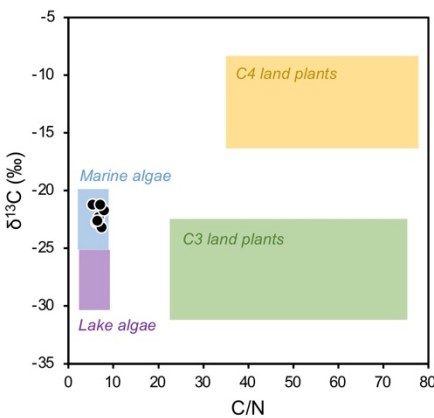


**Figure 3: Organic matter provenance based on bulk C/N and δ¹³C values. Reference values for four end-members after Meyers (1994).**

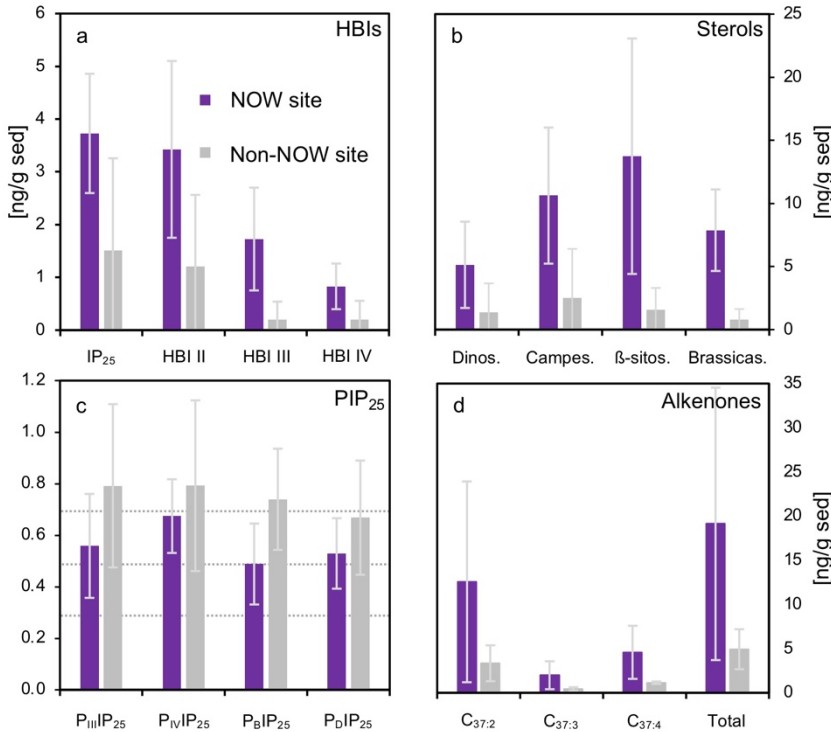

**Figure 4: Average concentrations sea ice and productivity biomarkers: a) HBIs, b) sterols, c) PIP₂₅ indices, and d) alkenones. Standard deviations for each shown in light gray. Sites within the NOW are colored purple whereas sites outside the NOW are gray. In panel c, dotted lines delimit qualitative sea ice concentration limits after Müller et al. (2011) for no sea ice (0 to 0.3), reduced sea ice (0.3 to 0.5), seasonal sea ice (0.5 to 0.7), and perennial sea ice (0.7 to 1).**




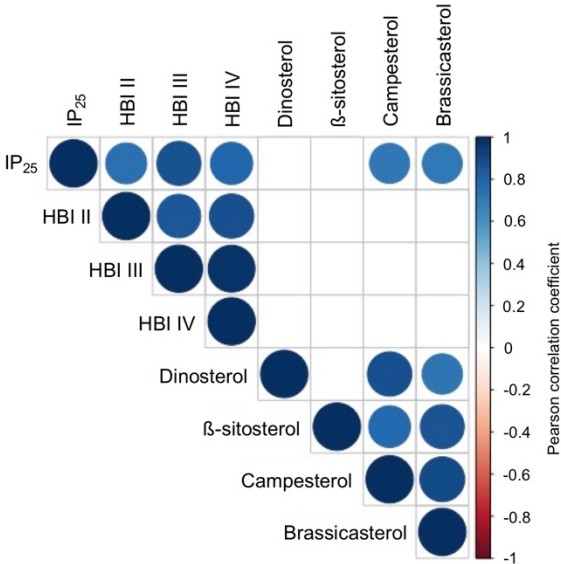

**Figure 5: Pearson correlation coefficients between HBIs and sterols. Positive correlations are displayed in blue and negative correlations in red. Both color and the size of the circle are proportional to the correlation coefficients. Insignificant correlations ($p$-values >0.01) are left blank.**


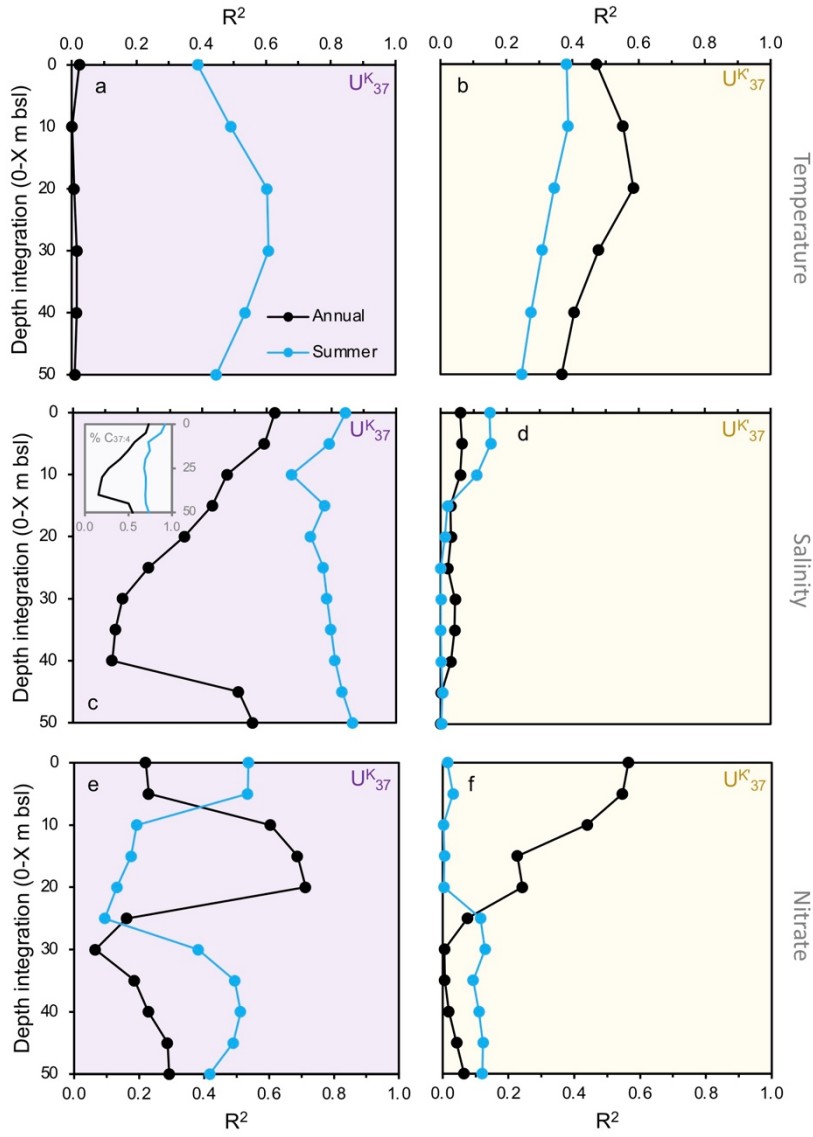

**Figure 6: Regression coefficients of alkenone-based indices against WOA18 temperature, salinity and depth at various depth integrations and seasons. a) $U^K_{37}$ and temperature, b) $U^{K'}_{37}$ and temperature, c) $U^K_{37}$ and salinity, inset: $C_{37:4}$ and salinity, d) $U^{K'}_{37}$ and salinity, e) $U^K_{37}$ and nitrate, and f) $U^{K'}_{37}$ and nitrate. WOA18 data from Garcia et al. (2018b), Locarnini et al. (2018) and Zweng et al. (2018).**


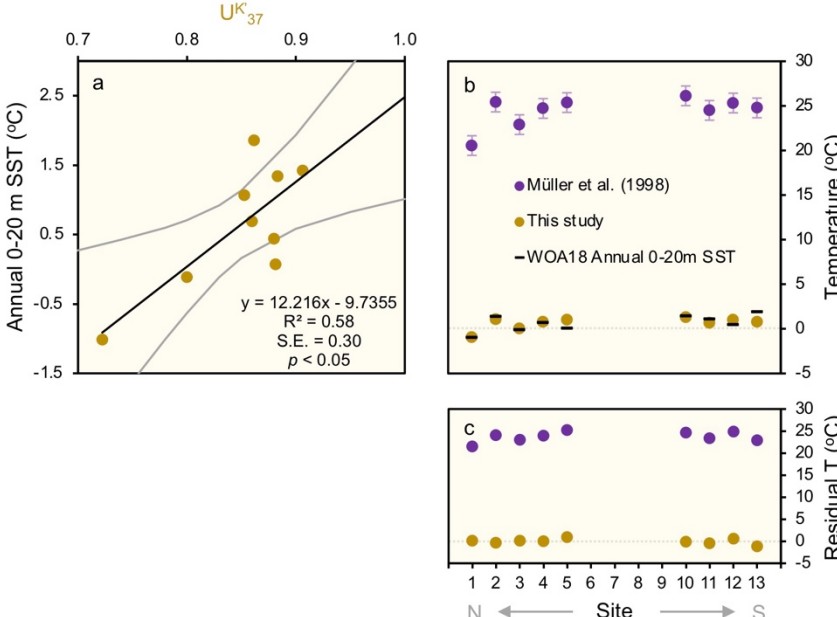

**Figure 7: Alkenone temperature and salinity calibrations. Shown are a) annual U$^{K'}_{37}$ 0-20m SST temperature calibration, b) U$^{K'}_{37}$ SST estimates based on calibrations from global (yellow, Müller et al., 1998), and Baffin Bay (purple, this study) against WOA18 annual 0-20m SST (black dashes, Locarnini et al., 2018), c) residuals of calibration estimates and WOA18 data. In panel d, the x-axis refers to site # as shown in Fig. 1 and Table 1. Sites 6-9 did not contain alkenones.**





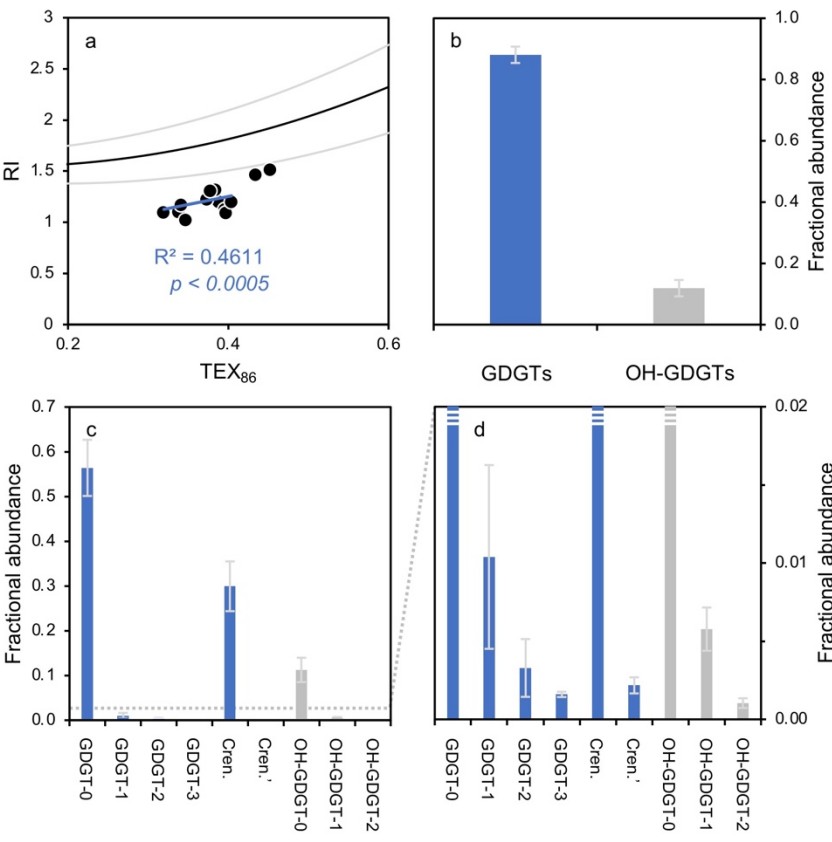

**Figure 8: GDGT- and OH-GDGT information. a) TEX$_{86}$ versus RI showing the global polynomial equation and 95% uncertainty envelope (black and gray lines, Zhang et al., 2016) and Baffin Bay sediments (black dots), b) fractional abundance of total GDGTs and OH-GDGTs, c) and d) fractional abundance of individual GDGTs (blue) and OH-GDGTs (gray) at two different scales.**


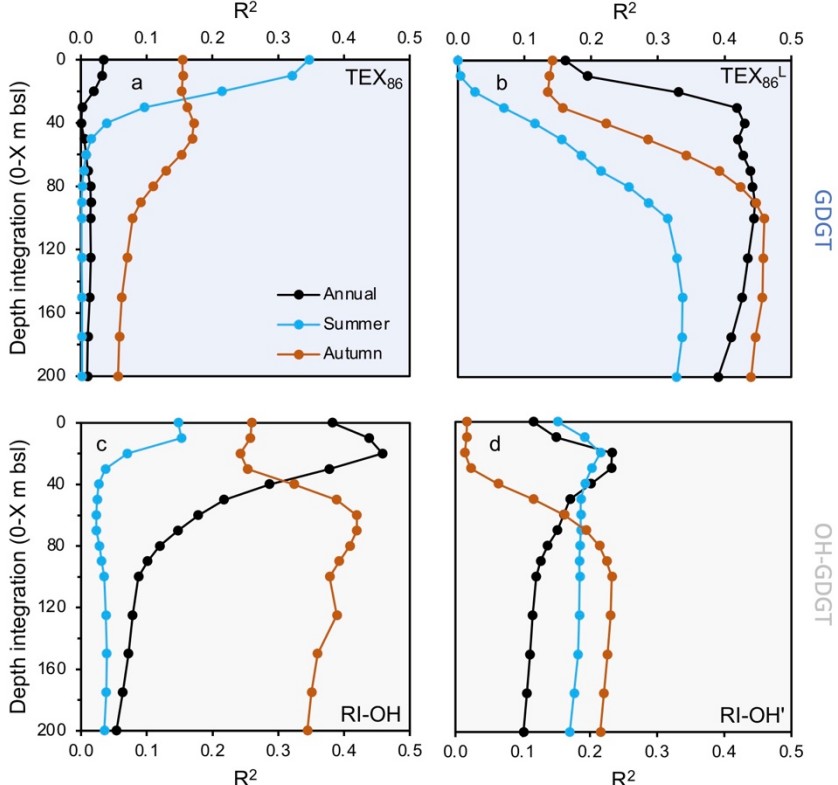

**Figure 9: Regression coefficients of GDGT-based temperature indices against WOA18 temperature at various depth integrations and seasons. a) TEX$_{86}$, b) TEX$_{86}^{L}$, c) RI-OH, and d) RI-OH'. WOA18 data from Locarnini et al. (2018).**


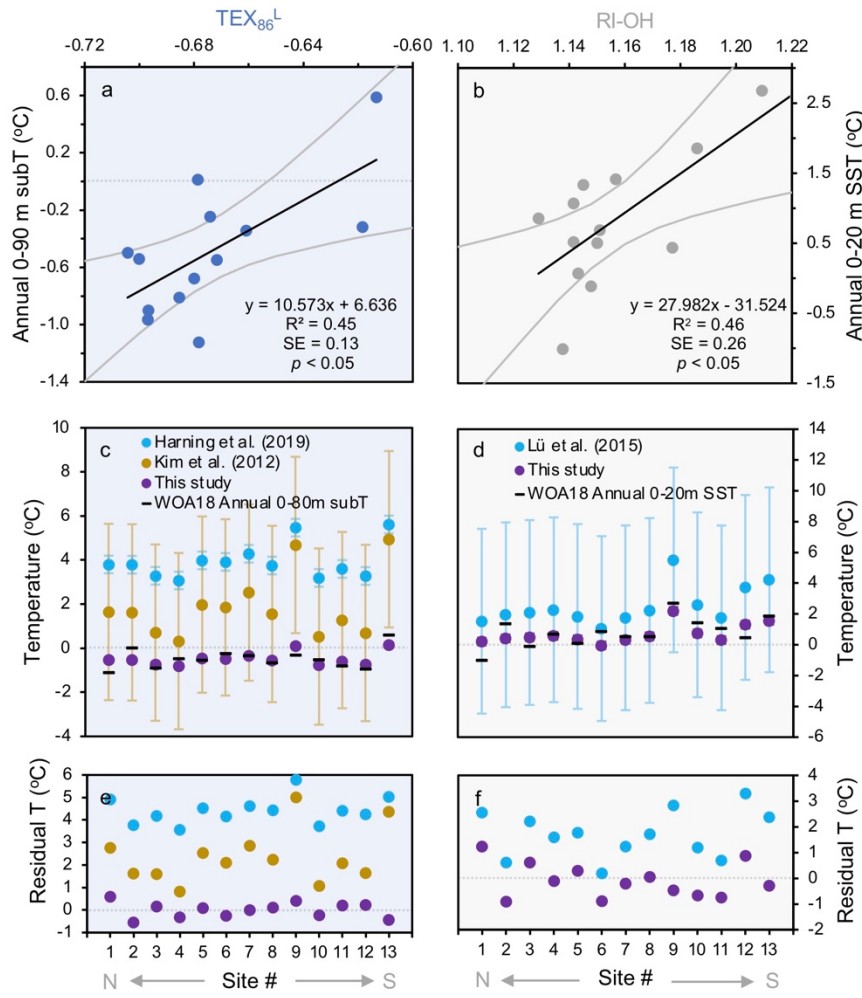

**Figure 10: GDGT- and OH-GDGT-temperature calibrations.** Shown are a) $TEX_{86}^L$ vs annual 0-90 m subT, b) RI-OH (OH-GDGTs) vs annual 0-20 m SST, c) $TEX_{86}^L$ subT estimates based on calibrations from Iceland (blue, Harning et al., 2019), global (yellow, Kim et al., 2012), and Baffin Bay (purple, this study) against WOA18 annual 0-90m subT (black dashes, Locarnini et al., 2018), d) RI-OH SST estimates based on calibrations from global (blue, Lü et al., 2015) and Baffin Bay (purple, this study) against WOA18 annual 0-20m SST (black dashes, Locarnini et al., 2018). Panels e and f show residuals of calibration estimates and WOA18 data for $TEX_{86}^L$ and RI-OH, respectively. The x-axis refers to site # as shown in Fig. 1 and Table 1.






**Table 1: Marine surface sediment site information.**

| Site # | Core Site Name | Lat | Long | Water depth (m bsl) |
|---|---|---|---|---|
| 1 | AMD17-129-BC | 78.42 | -74.24 | 521 |
| 2 | AMD17-101-BC | 76.48 | -77.77 | 378 |
| 3 | AMD17-108-BC | 76.47 | -74.70 | 449 |
| 4 | AMD17-111-BC | 76.44 | -73.32 | 593 |
| 5 | AMD17-115-BC | 76.57 | -71.33 | 668 |
| 6 | HU2009 029-040 BX | 75.58 | -78.63 | 580 |
| 7 | HU2008 029-59 TC | 74.26 | -82.23 | 791 |
| 8 | HU2008 029-49 BX | 74.03 | -77.13 | 868 |
| 9 | AMD17-BB2-BC | 72.77 | -67.25 | 2373 |
| 10 | AMD15-CASQ1-BC4 | 71.41 | -70.89 | 702 |
| 11 | AMD17-176-BC | 69.82 | -65.46 | 267 |
| 12 | AMD17-Site 8.1-BC | 69.52 | -64.97 | 1054 |
| 13 | AMD17-Disko Fan-BC | 67.99 | -59.60 | 1012 |