# Peer review of "Biomarker characterization of the North Water Polynya, Baffin Bay: Implications for local sea ice and temperature proxies"

_Biogeosciences, 2021_

## Referee Comment (RC2)

bg-2021-177    Submitted on 06 Jul 2021
**Biomarker characterization of the North Water Polynya, Baffin Bay:
Implications for local sea ice and temperature proxies**
David J. Harning, Brooke Holman, Lineke Woelders, Anne E. Jennings, and Julio
Sepúlveda

**Review by R. Stein (MARUM, Bremen University & AWI Bremerhaven)**

The very well paper by Harning et al. is dealing with specific biomarkers determined in
surface sediments from northern Baffin Bay, with a special focus on the North Water
Polynya. These biomarkers, i.e., highly-branched isoprenoids (HBIs), sterols, long-chain
alkenones and archaeal GDGTs, may give information about sea-ice conditions, open-water
productivity, ocean temperature, and terrigenous input, as shown in many previous studies
from very different ocean regions. Although major progress in using these proxies for
reconstruction of present and past environmental conditions has been obtained, more ground
truth data are still needed to fully approve some of these proxies, especially the proxies
dealing with the reconstruction of sea ice (for example see reviews by Stein et al., 2012; Belt
and Müller, 2013; Belt, 2018). Gaps in knowledge are related to the definite identification of
the source of specific biomarkers, (regional) proxy calibration to allow (semi)quantitative
estimates of sea-ice extent and its seasonal variability, sea-surface temperature, salinity etc..
The missing ground truth data can be obtained by detailed studies of sediment trap material
and surface sediments as well as cultural experiments, and are especially needed from the
high (polar) latitudes, e.g., the Arctic Ocean and its marginal seas.

In this context, the new data set from Harning et al. may give some important insight for
using these biomarkers for characterizing the modern environmental conditions in a large
polynya setting, i.e., an area of reduced sea ice and increased primary productivity. Such
data might be strongly relevant for using the proxies for reconstruction of the development of
past polynyas. Furthermore, Harning's data set allows to directly correlate a large variety of
biomarkers (often either HBI and sterol data are produced/published from the same set of
samples or alkenones or GDGTs). Despite this positive aspect, however, I have major
problems with the present version of the manuscript. These concerns are related to the data
set itself, the postulation of new SST calibrations for high latitudes based on a very limited
data set, the missing more detailed comparison/discussion of the own data with the
published biomarker data from Kolling et al. (2020), and some statements related to the
interpretation of the biomarker data in terms of their origin (i.e., marine vs. terrigenous), as
outlined in the following paragraphs. In my mind, a major clarification and revision of the
manuscript is needed before it can be accepted for publication.

   (1) Data base and presentation of data

The total number of data points (13 samples in total) is quite limited for the quite general
statements and interpretation of data given here. Whereas for the polynya itself (eight
samples) this might be ok, it is quite questionable for the area outside the polynya (five
samples for the entire northern Baffin Bay). Furthermore, these five samples are from very
different settings (one from the central Baffin Bay, close to Davis Strait, and three off different
fjord mouths), i.e., areas with very different sea-ice conditions and sedimentation rates. Due
to the latter, these 1 cm thick samples (proxy data) might represent quite different time
intervals (average sea-ice condition, temperatures etc. over 10 to hundreds of years).
Furthermore, a presentation of the data in terms of distribution maps (see Kolling et al.,
2020) might be useful here as well. In order to allow more general
statements/interpretations/conclusions, the discussion of the new data together with the data
from Kolling et al. (2020) would be most helpful/important. Kolling et al. (2020) have studied
samples from locations very close to those of this study (Fig. R1). Thus, data from the same
biomarkers (i.e., IP25, HBIs and sterols as well as the different PIP25) are available for a

detailed comparison and interpretation in terms of sea-ice extent and its seasonality, productivity etc. (see examples in Fig. R1).

(2) Low concentrations of HBIs and sterols

When I myself tried to roughly compare the new Harning et al. data with our Kolling et al. (2020) data, I realized a "problem", and this is one of my major concerns I have with the new data presented here. The absolute concentrations of the HBIs and the sterols are significantly lower (two orders of magnitude or so!!) than those presented by Kolling et al. (2020). What might be the cause for this? Storage of samples (fresh/deep-frozen samples vs. samples stored under room temperature), or different analytical approach? I am not a chemist. Thus, I myself cannot comment in detail the analytical approach for identification and quantification of the biomarkers that has been used here, but I know from cooperation with the chemists involved in our biomarker analyses as well as the Simon Belt group that the analytical procedure is overall significance if data from different labs will be compared. Thus, Belt et al. (2014) carried-out an inter-laboratory investigation dealing with the identification and quantification of the Arctic sea ice biomarker proxy IP25 and other HBIs in marine sediments (our lab was involved in this study as well). As final statements these authors summarized in their abstract that "data are presented that suggest that extraction of IP25 is consistent between Automated Solvent Extraction (ASE) and sonication methods and that IP25 concentrations based on 7-hexylnonadecane as an internal standard are comparable using these methods. Recoveries of some more unsaturated HBIs and the internal standard 9-octylheptadecene, however, were lower with the ASE procedure, possibly due to partial degradation of these more reactive chemicals as a result of higher temperatures employed with this method. For future measurements, we recommend the use of reference sediment material with known concentration(s) of IP25 for determining and routinely monitoring instrumental response factors."

Harning et al. have used the ASE for extraction, and they should check their HBI and sterol data and comment on their analytical approach. For example, did you take in account the different response factors for the analytes and internal standards?

(3) Alkenones, high UK37', and new SST calibration

Concerning the alkenone data, I also have problems. The northern Baffin Bay is an environment with sea-surface temperatures significantly lower than 10°C (<5°C). Under such cold conditions, the C37:3 long-chain alkenones should be predominant over the C37-2 alkenones (e.g., Prahl and Wakeham, 1987). In this study, however, the C37:2 alkenones are predominant (Fig. 4 of their paper), resulting in high UK37' values of 0.7-0.9 (Fig. 7 of their paper), i.e., values that are typical for much higher temperatures. In polar and subpolar regions including Baffin Bay (!), however, UK37' values are typically between 0.2 and 0.4 (e.g., Rosell-Melé, 1998; Bendle and Rosell-Melé, 2004; Méheust et al., 2013; Moros et al., 2016). The authors are aware about this problem and discuss that temperature, salinity and nitrate are factors that may have influenced UK37' and UK37 values. Nevertheless, they state that "given our limited dataset at this time, temperature seems to be the most important environment variable on UK'37 values", and then simply have correlated the high UK37' values with the measured low SST values of Baffin Bay. As result, they obtain, of course and not surprising, a new calibration that give totally different SST values in comparison to those calculated based on Müller et al. (1998) (Fig. 7). Did the authors test their "new calibration" using the UK37' data from Moros et al. (2016), i.e., data from Baffin Bay? What SST values they would get? Furthermore, there might be another option, i.e., there might be some question mark related the data that should be clarified before postulating a new calibration (e.g., what about co-elution with other compounds that might result in too high C37:2 concentrations; see Villanueva and Grimmalt; 1996). Please check!

In the introduction (Lines 55-57), some credit should be given to other studies dealing with the use and calibration of UK37 and TEX86 for SST reconstructions in high latitudes (Sikes et al., 1997; Rosell-Melé, 1998; Ho et al., 2014)

**(4) Biomarkers (sterols) and their sources**

I do not agree with the general statement that in the Arctic the abundance of brassicasterol, dinosterol, campesterol and ß-sitosterol is mainly related to marine productivity. The interpretation of the sterols and their use as organic-carbon source indicator are not easy tasks and may strongly differ from region to region (e.g., Volkman, 1986; Huang and Meinschein, 1976; Fahl and Stein, 1997, 1999; Belt et al., 2013). Brassicasterol often used as "marine productivity indicator" might be ok in areas not influenced by strong terrigenous input (river discharge). In coastal areas controlled by huge river discharge, such as the Kara and Laptev seas, a large amount of brassicasterol found in surface sediments have a terrestrial (lacustrine) source (e.g., Fahl et al., 2003; Hörner et al., 2016). Furthermore, in these shallow-water coastal zones, terrestrial/lacustrine brassicasterol as well as brassicasterol produced by marine algae may be incorporated into sea ice and transported into the open central Arctic Ocean. Thus, these biomarkers not produced by sea ice may be found in the sea ice and result in erroneous interpretation of the source of this biomarker. When using brassicasterol as "open-water productivity proxy", additional information about the environmental situation should be taken into account and additional biomarkers (such as dinosterol and the HBI-III) should be used as well. When using the PIP25 approach for reconstructing sea-ice conditions, PIP25 values based on IP25 and brassicasterol, dinosterol and HBI-III, should be calculated and discussed.

Vascular plants are producers of campesterol and ß-sitosterol (Huang and Meinschein, 1976) but may also be produced by diatoms (Belt et al., 2013). In the Arctic Ocean characterized by strong terrigenous input into the marginal sea and – via sea ice and ocean currents – a predominantly terrigenous source of these biomarkers is most probable (e.g., Yunker et al., 1995, 2005; Stein and Macdonald, 2004; Xiao et al., 2013). Such riverine input of organic carbon onto the shelf is nicely reflected in maximum concentration of campesterol and ß-sitosterol in surface sediments close to the major river mouths in the Kara and Laptev seas (Xiao et al., 2013; Fig. R2). In most part of the Arctic Ocean, thus terrigenous organic matter is predominant in surface sediments (Stein and Macdonald, 2004; Fig. R2). Thus, I. cannot agree with the authors' statement in 113/114): "In the Arctic where terrestrial biomass is low, we assume that the contribution of terrestrial-derived campesterol and β-sitosterol is minimal compared to that produced in the ocean."

References
Belt, S. T., 2018. Source-specific biomarkers as proxies for Arctic and Antarctic sea ice. Org. Geochem., 125, 277-298.
Belt, S. T., and Müller, J., 2013. The Arctic sea ice biomarker IP25: A review of current understanding, recommendations for future research and applications in palaeo sea ice reconstructions. Quaternary Science Reviews, 79, 9–25. https://doi.org/10.1016/j. quascirev.2012.12.001
Belt, S. T., Brown, T. A., Ampel, L., Cabedo-Sanz, P., Fahl, K., Kocis, J. J., & Xu, Y., 2014. An inter-laboratory investigation of the Arctic sea ice biomarker proxy IP25 in marine sediments: Key outcomes and recommendations. Climate of the Past, 10(1), 155–166. https://doi.org/10.5194/cp-10-155-2014
Belt, S. T., Brown, T. A., Ringrose, A. E., Cabedo-Sanz, P., Mundy, C. J., Gosselin, M., and Poulin, M., 2013. Quantitative measurement of the sea ice diatom biomarker IP25 and sterols in Arctic sea ice and underlying sediments: further considerations for palaeo sea ice reconstruction. Org. Geochem., 62, 33-45.
Bendle, J., and A. Rosell-Melé, 2004. Distributions of UK37 and UK37' in the surface waters and sediments of the Nordic Seas: Implications for paleoceanography. Geochem. Geophys. Geosyst. 5, Q11013, doi:10.1029/2004GC000741.
Fahl, K. and Stein, R., 1997. Modern organic-carbon-deposition in the Laptev Sea and the adjacent continental slope: Surface-water productivity vs. terrigenous input. Org. Geochem., 26: 379-390.

Fahl, K. and Stein, R., 1999. Biomarkers as organic-carbon-source and environmental indicators in the Late Quaternary Arctic Ocean: Problems and perspectives. Mar. Chem., 63/3-4: 293-309.

Fahl, K. and Stein, R., 2012. Modern seasonal variability and deglacial/Holocene change of central Arctic Ocean sea-ice cover: New insights from biomarker proxy records. Earth Planet. Sci. Lett. 351-352C , 123-133;  doi:10.1016/j.epsl.2012.07.009.

Fahl, K., Stein, R., 1997. Modern organic carbon deposition in the Laptev Sea and the adjacent continental slope: Surface-water productivity vs. terrigenous input. Organic Geochemistry 26, 379-390.

Fahl, K., Stein, R., Gaye-Haake, B., Gebhardt, C., Kodina, L.A., Unger, D., and Ittekkot, V., 2003. Biomarkers in surface sediments from  Ob and Yenisei estuaries and southern Kara Sea: Evidence for particulate organic carbon sources, pathways, and degradation. In: Stein, R., Fahl, K., Fütterer, D.K., Galimov, E. M., and Stepanets, O.V. (Eds.), Siberian River Run-off in the Kara Sea: Characterisation, Quantification, Variability, and Environmental Significance, Proceedings in Marine Sciences Vol.6, Elsevier, Amsterdam, 329-348.

Ho, S.L., Mollenhauer, G., Fietz, S., Martinez-Garcia, A., Lamy, F., Rueda, G., Schipper, K., Méheust, M., Rosell-Melé, A., Stein, R., Tiedemann, R., 2014. Appraisal of the $TEX_{86}$ and $TEX_{86}^L$ thermometries in the subpolar and polar regions. Geochimica et Cosmochimica Acta, 131 , pp. 213-226 . doi: 10.1016/j.gca.2014.01.001.

Hörner, T., Stein, R., Fahl, K., Birgel, D., 2016. Post-glacial variability of sea ice cover, river run-off and biological production in the western Laptev Sea (Arctic Ocean) - A high-resolution biomarker study. Quaternary Science Reviews 143, 133-149. http://dx.doi.org/10.1016/j.quascirev.2016.04.011

Huang, W. Y., and Meinschein, W. G., 1976.  Sterols as source indicators of organic materials in sediments. Geochem. Cosmochim. Acta 40, 323-330.

Kolling, H. M., Stein, R., Fahl, K., Sadatzki, H., de Vernal, A., Xiao, X., 2020. Biomarker distributions in(sub)-Arctic surface sediments andtheir potential for sea icereconstructions. Geochemistry, Geophysics, Geosystems 21, e2019GC00862. https://doi.org/10.1029/2019GC008629

Méheust, M., Fahl, K., Stein, R., 2013. Variability in modern sea-surface temperature, sea ice and terrigenous input in the sub-polar North Pacific and Bering Sea: Reconstruction from biomarker data. Org. Geochemistry 57, 54-64.

Moros, M., Lloyd, J.M., Perner, K., Krawczyk, D., Blanz, T., de Vernal, A., Ouellet-Bernier, M.-M., Kuijpers, A., Jennings, A.E., Witkowski, A., Schneider, R., Jansen, E., 2016. Surface and sub-surface multi-proxy reconstruction of middle to late Holocene palaeoceanographic changes in Disko Bugt, West Greenland. Quaternary Science Reviews 132, 146-160.

Müller, P. J., Kirst, G., Ruhland, G., von Storch, I., and Rosell-Mel., A., 1998. Calibration of the alkenone paleotemperature index UK'37 based on core-tops from the eastern South Atlantic and the global ocean (60°N-60°S). Geochim. Cosmochim. Acta 62, 1757-1772, 1998.

Prahl, F. G., and Wakeham, S. G., 1987. Calibration of unsaturation patterns in long-chain ketone compositions for paleotemperature assessment. Nature, 330, 533-537.

Rosell-Melé, A., 1998. Interhemispheric appraisal of the value of alkenone indices as temperature and salinity proxies in high latitude locations. Paleoceanography 13, 694.

Sikes, E.L., Volkman, J.K., Robertson, L.G., Pichon, J.-J., 1997. Alkenones and alkenes in surface waters and sediments of the Southern Ocean: Implications for paleotemperature estimation in polar regions. Geochimica et Cosmochimica Acta 61, 1495-1505

Stein, R. and Macdonald, R.W. (Eds.), 2004. The Organic Carbon Cycle in the Arctic Ocean, Springer-Verlag, Berlin, 363 pp.

Stein, R., 2008. Arctic Ocean Sediments: Processes, Proxies, and Palaeoenvironment. Developments in Marine Geology, Vol. 2, Elsevier, Amsterdam, 587 pp..

Stein, R., Fahl, K., and Müller, J., 2012. Proxy reconstruction of Arctic Ocean sea ice history: „From IRD to $IP_{25}$". Polarforschung 82, 37-71, hdl:10013/epic.40432.d001

Villanueva, J. and Grimmalt, J.O., 1996. Pitfalls in the chromatographic determination of the alkenone UK37 index for paleotemperature estimation. Journal of Chromatography A 723, 285-291.

Volkman, J. K.,1986. A review of sterol markers for marine and terrigenous organic matter. Org. Geochem., 9, 83-99.

Xiao, X., Fahl, K., and Stein, R., 2013.  Biomarker distributions in surface sediments from the Kara and Laptev Seas (Arctic Ocean): Indicators for organic-carbon sources and sea-ice coverage. Quat. Sci. Rev., doi.org/10.1016/j.quascirev.2012.11.028.

Yunker, M.B., Belicka, L.L., Harvey, H.R., Macdonald, R.W., 2005. Tracing the inputs and fate of marine and terrigenous organic matter in Arctic Ocean sediments: a multivariate analysis of lipid biomarkers. Deep-Sea Research II 52, 3478-3508.

Yunker, M.B., Macdonald, R.W., Veltkamp, D.J., Cretney, W.J., 1995. Terrestrial and marine biomarkers in a seasonally ice-covered Arctic estuary-integration of multivariate and biomarker approaches. Marine Chemistry 49, 1-50.

*Correlation Coefficient (r) Between PIP$_{25}$ Indices With Sea Ice Concentrations From Baffin Bay for Different Seasons*

| Season | P$_B$IP$_{25}$ | P$_D$IP$_{25}$ | P$_{III}$IP$_{25}$ |
|--------|--------|--------|--------|
| Spring | 0.80 | 0.76 | 0.63 |
| Summer | 0.55 | 0.55 | 0.28 |
| Autumn | 0.83 | 0.85 | 0.57 |
| Winter | 0.79 | 0.77 | 0.58 |

[Figure]

Fig. R1. Comparison of sea ice reconstructions using PDIP25 and PIIIP25 with the seasonal sea ice distribution in Baffin Bay (Kolling et al., 2020 with further references therein). In the maps with modern sea-ice distribution, the locations of the surface sediment samples of this study are shown as white circles.

[Figure]

Fig. R2

A. Terrigenous (TOM) and marine (MOM) organic carbon in surface sediments from Arctic marginal seas and central Arctic Ocean and main processes controlling organic carbon (OC) input. In the pie diagrams, average percentage values of terrigenous (brown colour) and marine, i.e., marine/aquatic and ice algae (blue colour), are given. In addition, OC accumulation rates for the Arctic marginal seas (bold numbers in brackets) are listed. For data base see Stein & Macdonald (2004), Stein (2008), and references therein. Colour bar in the right indicates average summer sea-ice concentrations (1988-2007) shown in the map (Source: Stein, 2017).

B. Concentrations (µg/g TOC) of terrigenous biomarkers (campesterol and ß-sitosterol) in surface sediments from the Kara and Laptev seas (Source: Xiao et al., 2013).

---

## Author Comment (AC1)

Reviewer 1:

David Harning and colleagues present new biomarker data from 13 surface sediment samples in northern and western Baffin Bay, focusing on the region of the North Water Polynya. Biomarker data comprise HBIs, sterols, alkenones, and GDGTs, representing a comprehensive study of the use of biomarker-based sea-ice and temperature proxies in Baffin Bay and for the characterization of polynya dynamics. They conclude that the pelagic and sympagic productivity is an order of magnitude higher at the polynya sites, compared to sites outside of the NOW region. This translates to low (higher) $PIP_{25}$ indices within (outside) of the NOW, in line with satellite-derived sea-ice concentration. Harding et al. recommend using sterols rather than HBI III to calculate $PIP_{25}$ indices in the Baffin Bay region, due to uncertainty with regard to HBI III producing species. Further, they propose regional temperature calibrations for both alkenone and GDGT indices, but also show that other environmental variables might contribute to the variability in alkenone and GDGT assemblage recorded in sediments.

The manuscript is very well written and follows a clear storyline with a logical succession of scientific arguments. The presented data fits within the remit of Biogeosciences and I only have a few minor comments regarding comparison with previously published data, and correlations of the environmental and biomarker data. Thus, I recommend publication in Biogeosciences following minor revisions.

> We kindly thank Reviewer 1 for their thoughtful review of our manuscript and offering valuable suggestions to improve its overall quality. Below we provide responses to each of their comments and suggestions and look forward to submitting a revised and stronger manuscript.

General comments:

1. Please comment on why the concentrations of HBIs in the Baffin Bay surface sediments (in part from the same samples) vary by orders of magnitude between the data presented here and the data presented in Kolling et al. 2020.

   > Please see our detailed reply to a similar comment by Reviewer 2. We plan to double check our quantification approach using synthetic standards that are structurally more similar to HBIs, which have been requested from colleagues.

2. The Baffin Bay data presented in Kolling et al. 2020 do not show lower $P_BIP_{25}$ and $P_DIP_{25}$ indices in the NOW region. Please comment on this in the manuscript. Is it possible to integrate your data with previously published data to strengthen/confirm the arguments made?

   > Given that we may be underestimating the concentration of sterols (please see detailed reply to Reviewer 2), the values of $P_BIP_{25}$ and $P_DIP_{25}$ are subject to change as we test different standards. Therefore, we refrain from making any conclusions about the similarity or lack thereof between the PIP indices of Kolling et al. (2020) and those in our manuscript at this time. However, once our data is finalized, we will make more detailed and explicit comparisons between the two datasets in the revised manuscript.

3. For the data presented in Figure 6 and 9:

    1. Why are alkenone indices not compared to autumn conditions?

    This was an oversight in the data comparison, and we appreciate the notice. We will add a comparison of alkenones with autumn conditions in the revised manuscript.

    2. What is the significance level for the presented $R^2$ values? Can you include confidence intervals?

    Calculating $p$ values is a common method for determining the significance of a statistical tests, which we will assess for all $R^2$ values in the manuscript (i.e., correlations between alkenones and GDGTs and the corresponding WOA18 environmental variables).

    3. I also wonder how the uncertainty of the WOA18 data for each given depth interval and station influences the significance of $R^2$. However, I am unsure of the best way to test this, maybe you could consider determination of $R^2$ confidence intervals using bootstrapping where the environmental data (e.g. temperature) is based on random sampling of normal distributions characterised by the mean and standard deviation of the data for a given depth interval and station.

    This is an excellent point by the reviewer and nicely follows the previous comment. In addition to $p$ values, performing bootstrap resampling would allow us to calculate confidence intervals for our $R^2$ values. We plan to take this advice and perform these analyses with our datasets.

    4. Considering the fragmentary availability of winter and spring WOA18 data, do the data summed up in the annual datasets (Fig. 6 and Fig. 9) cover the same interval for each year or is it an average of all data available for a given year? If the latter, does it make a difference if the data is restricted to the same seasons for every year?

    This is another excellent point by the reviewer, and we are glad to have the opportunity to expand upon the dataset. The annual WOA18 datasets are an average of the data available for the 12 months, so yes, the annual value would be influenced by how complete the given year is. In this regard, we do note in L236-238 that the annual data is more reflective of the ice-free months as winter data is unavailable due to seasonal ice cover. However, we can expand this sentence in the main text to also state that the availability of monthly data during the winter and spring months will ultimately impact the annual mean and introduce an added uncertainty when comparing annual correlations.

4. Line 459-460: This is a somewhat circular argument, as the temperature calibration is based on correlation with the WOA18 data.

    We agree with the reviewer and appreciate this highlight. We will remove this sentence in the revised version of the manuscript.

Visual presentation:

- In Figure 1, could you highlight the samples from where bulk geochemical proxies are available (e.g. differing fill/line colour)?

- Figures S1-S3: Please add the same style of overview maps for $C_{37:2}$, $C_{37:3}$, and $C_{37:4}$ to the supplementary information, so the reader can see which samples had alkenone concentrations over the limit of detection.

All figures will be edited accordingly, thank you for the suggestions.

Typographic comments:

- Line 25: Remove bvc from Kuhlbrodt et al., 2017

- Check spelling of sea ice vs. sea-ice for consistency. The dominantly used spelling is sea ice, but sea-ice is used in 4 instances (line 45, 63, 98, 436).

- Lines 73-74: 'In contrast, the NOW has anomalously low concentrations of thin ice, even during winter months.' Maybe rephrase this sentence for clarity (anomalously low concentrations of thin ice can also be read as meaning thick sea ice).

- Line 204: There is a full stop missing after '…for 20 min.)'

- Lines 501-502: check nitrate vs. nitrite

- Revise formatting of the reference list, to comply with the style of Biogeosciences.

All typographic comments will be corrected, thank you.

---

## Author Comment (AC2)

**Reviewer 2 (Ruediger Stein):**

The very well paper by Harning et al. is dealing with specific biomarkers determined in surface sediments from northern Baffin Bay, with a special focus on the North Water Polynya. These biomarkers, i.e., highly-branched isoprenoids (HBIs), sterols, long-chain alkenones and archaeal GDGTs, may give information about sea-ice conditions, open-water productivity, ocean temperature, and terrigenous input, as shown in many previous studies from very different ocean regions. Although major progress in using these proxies for reconstruction of present and past environmental conditions has been obtained, more ground truth data are still needed to fully approve some of these proxies, especially the proxies dealing with the reconstruction of sea ice (for example see reviews by Stein et al., 2012; Belt and Müller, 2013; Belt, 2018). Gaps in knowledge are related to the definite identification of the source of specific biomarkers, (regional) proxy calibration to allow (semi)quantitative estimates of sea-ice extent and its seasonal variability, sea-surface temperature, salinity etc. The missing ground truth data can be obtained by detailed studies of sediment trap material and surface sediments as well as cultural experiments and are especially needed from the high (polar) latitudes, e.g., the Arctic Ocean and its marginal seas.

In this context, the new data set from Harning et al. may give some important insight for using these biomarkers for characterizing the modern environmental conditions in a large polynya setting, i.e., an area of reduced sea ice and increased primary productivity. Such data might be strongly relevant for using the proxies for reconstruction of the development of past polynyas. Furthermore, Harning's data set allows to directly correlate a large variety of biomarkers (often either HBI and sterol data are produced/published from the same set of samples or alkenones or GDGTs). Despite this positive aspect, however, I have major problems with the present version of the manuscript. These concerns are related to the data set itself, the postulation of new SST calibrations for high latitudes based on a very limited data set, the missing more detailed comparison/discussion of the own data with the published biomarker data in terms of their origin (i.e., marine vs. terrigenous), as outlined in the following paragraphs. In my mind, a major clarification and revision of the manuscript is needed before it can be accepted for publication.

We kindly thank Ruediger Stein (Reviewer 2) for his detailed review of our manuscript and highlighting some key areas for improvement. Below we provide responses to each of his comments and suggestions and look forward to submitting a revised and stronger manuscript.

**Data base and presentation of data:**

The total number of data points (13 samples in total) is quite limited for the quite general statements and interpretation of data given here. Whereas for the polynya itself (eight samples) this might be ok, it is quite questionable for the area outside the polynya (five samples for the entire northern Baffin Bay). Furthermore, these five samples are from very different settings (one from the central Baffin Bay, close to Davis Strait, and three off different fjord mouths), i.e., areas with very different sea-ice conditions and sedimentation rates. Due to the latter, these 1 cm thick samples (proxy data) might represent quite different time intervals (average sea-ice condition, temperatures etc. over 10 to hundreds of years). Furthermore, a presentation of the data in terms of distribution maps (see Kolling et al., 2020) might be useful here as well. In order

to allow more general statements/interpretations/conclusions, the discussion of the new data together with the data from Kolling et al. (2020) would be most helpful/important. Kolling et al. (2020) have studied samples from locations very close to those of this study (Fig. R1). Thus, data from the same biomarkers (i.e., IP25, HBIs and sterols as well as the different PIP25) are available for a detailed comparison and interpretation in terms of sea-ice extent and its seasonality, productivity etc. (see examples in Fig. R1).

While we are fully aware of the limited size of our dataset, it was what was made available for our study, especially in the polynya region where we plan to apply these calibrations for paleoceanographic reconstructions. The reviewer is certainly correct that the different locations and environments of the samples can lead to different integrated intervals of time, which we currently highlight in L159 of the manuscript. However, this is an issue for any surface sediment proxy calibration that covers regions with differing sedimentation rates and will always introduce added uncertainty. By acknowledging this limitation as we currently have, we hope this satisfies the reviewer's concern.

We do currently provide distribution maps for the HBIs and sterols (please see the supplemental material), however, we prefer not to interpolate the data as done in Kolling et al. (2020) to avoid over-interpretations related to spatial gaps in the datasets. Following the additional analytical steps outlined in our response to the following comment, we indeed plan to make more direct and integrated comparisons with the Kolling et al. (2020) datasets. We appreciate the opportunity to expand on this in the revised manuscript.

**Low concentrations of HBIs and sterols:**

When I myself tried to roughly compare the new Harning et al. data with our Kolling et al. (2020) data, I realized a "problem", and this is one of my major concerns I have with the new data presented here. The absolute concentrations of the HBIs and the sterols are significantly lower (two orders of magnitude or so!!) than those presented by Kolling et al. (2020). What might be the cause for this? Storage of samples (fresh/deep-frozen samples vs. samples stored under room temperature), or different analytical approach? I am not a chemist. Thus, I myself cannot comment in detail the analytical approach for identification and quantification of the biomarkers that has been used here, but I know from cooperation with the chemists involved in our biomarker analyses as well as the Simon Belt group that the analytical procedure is overall significance if data from different labs will be compared. Thus, Belt et al. (2014) carried-out an inter-laboratory investigation dealing with the identification and quantification of the Arctic sea ice biomarker proxy IP25 and other HBIs in marine sediments (our lab was involved in this study as well). As final statements these authors summarized in their abstract that "data are presented that suggest that extraction of IP25 is consistent between Automated Solvent Extraction (ASE) and sonication methods and that IP25 concentrations based on 7-hexylnonadecane as an internal standard are comparable using these methods. Recoveries of some more unsaturated HBIs and the internal standard 9-octylheptadecene, however, were lower with the ASE procedure, possibly due to partial degradation of these more reactive chemicals as a result of higher temperatures employed with this method. For future measurements, we recommend the use of reference sediment material with known concentration(s) of IP25 for determining and routinely monitoring instrumental response factors."

Harning et al. have used the ASE for extraction, and they should check their HBI and sterol data and comment on their analytical approach. For example, did you take in account the different response factors for the analytes and internal standards?

We thank the reviewer for this comment, and for the details of previous work on extraction techniques and quantification biases. The reviewer is indeed correct that the difference in analytical approach and storage of samples can both lead to underestimation of biomarker concentrations (Belt et al., 2012; Cabedo-Sanz et al., 2016). We used a methylated alkane (3-methylheneicosane) that our lab normally uses for aliphatic hydrocarbons. Unfortunately, we were not fully aware of how different the response factor of this standard could be compared to the HBI standards used by Belt et al. (2012) and Kolling et al. (2020). We agree that the structural differences will lead to different ionization efficiencies and response factors, which can alter the calculation of HBI concentrations. In order to make our HBI datasets comparable to those of Kolling et al. (2020), we propose several steps that we will complete during the revision period:

- We have contacted the Belt lab to procure the 7-HND standard and will run a 7-HND dilution series under the same GC-MS operating conditions as we did for our samples to calculate its response factor. We will compare this against that of 3methylheneicosane.
- 2) The Belt lab has also provided us with sediment that contains a known concentration of the different HBIs that we will analyze on our GC-MS. We aim to quantify HBIs in this sample using response factors from 7-HND and 3-methylheneicosane,
- 3) By comparing the differences in response factors (step 1) with the peak areas identified by their molecular ions in SIM mode (step 2), we can correct our initial response factors to produce corrected concentrations.

While the standards used for sterol quantification are less standardized, we do note that Kolling et al. (2020) used a different standard for quantification in their study (cholesterol- $D_6$ ) than the one we used (1-nonadecanol). Therefore, to make our datasets more comparable, we will follow a similar approach as outlined above for HBIs. More specifically, we will generate dilution series for several sterols (cholesterol, ergosterol and stigmasterol). The cholesterol standard will allow us to correct our brassicasterol and dinosterol concentrations, both of which were analyzed by Kolling et al. (2020). The dilution series for ergosterol and stigmasterol (two phytosterols) will allow us to test if the ionization efficiencies in these structurally similar sterols is akin to cholesterol's. If they are indeed similar as expected, then we can adjust the concentrations for our campesterol and ß-sitosterol accordingly. However, Kolling et al. (2020) did not analyze these latter phytosterols, so no comparison can be made.

Finally, if our proposed approach to correct our HBI and sterol concentrations does not bring the two different studies into alignment, then we will have to consider sample storage and/or extraction procedure as additional possible factors. In this regard, we note that Kolling et al. (2020) stored their sediment samples in glass vials or plastic bags below -20°C. On the other hand, our sediment samples were stored in glass vials but at 4°C. Previous studies have noted that biomarkers, such as HBIs, can degrade faster at warmer temperatures, although this was more readily observed at room temperature (Cabedo-Sanz et al., 2016), and thus, may not be a concern in our study. The extraction method (ASE vs ultrasonication) may also result in preferential degradation of lipids but would not be able to be corrected for. If such is the case, the relative abundance of the HBIs should be comparable to those published Kolling et al. (2020), and comparison between the two datasets could be achieved accordingly.

Alkenones, high UK37', and new SST calibration:

Concerning the alkenone data, I also have problems. The northern Baffin Bay is an environment with sea-surface temperatures significantly lower than 10°C (

In the introduction (Lines 55-57), some credit should be given to other studies dealing with the use and calibration of UK37 and TEX86 for SST reconstructions in high latitudes (Sikes et al., 1997; Rosell-Melé, 1998; Ho et al., 2014)

We agree with the reviewer that citing these previous studies is important and will be included in the revised manuscript.

Biomarkers (sterols) and their sources:

I do not agree with the general statement that in the Arctic the abundance of brassicasterol, dinosterol, campesterol and ß-sitosterol is mainly related to marine productivity. The interpretation of the sterols and their use as organic-carbon source indicator are not easy tasks and may strongly differ from region to region (e.g., Volkman, 1986; Huang and Meinschein, 1976; Fahl and Stein, 1997, 1999; Belt et al., 2013). Brassicasterol often used as "marine productivity indicator" might be ok in areas not influenced by strong terrigenous input (river discharge). In coastal areas controlled by huge river discharge, such as the Kara and Laptev

seas, a large amount of brassicasterol found in surface sediments have a terrestrial (lacustrine) source (e.g., Fahl et al., 2003; Hörner et al., 2016). Furthermore, in these shallow-water coastal zones, terrestrial/lacustrine brassicasterol as well as brassicasterol produced by marine algae may be incorporated into sea ice and transported into the open central Arctic Ocean. Thus, these biomarkers not produced by sea ice may be found in the sea ice and result in erroneous interpretation of the source of this biomarker. When using brassicasterol as "open-water productivity proxy", additional information about the environmental situation should be taken into account and additional biomarkers (such as dinosterol and the HBI-III) should be used as well. When using the PIP25 approach for reconstructing sea-ice conditions, PIP25 values based on IP25 and brassicasterol, dinosterol and HBI-III, should be calculated and discussed.

Vascular plants are producers of campesterol and ß-sitosterol (Huang and Meinschein, 1976) but may also be produced by diatoms (Belt et al., 2013). In the Arctic Ocean characterized by strong terrigenous input into the marginal sea and – via sea ice and ocean currents – a predominantly terrigenous source of these biomarkers is most probable (e.g., Yunker et al., 1995, 2005; Stein and Macdonald, 2004; Xiao et al., 2013). Such riverine input of organic carbon onto the shelf is nicely reflected in maximum concentration of campesterol and ß-sitosterol in surface sediments close to the major river mouths in the Kara and Laptev seas (Xiao et al., 2013; Fig. R2). In most part of the Arctic Ocean, thus terrigenous organic matter is predominant in surface sediments (Stein and Macdonald, 2004; Fig. R2). Thus, I. cannot agree with the authors' statement in 113/114): "In the Arctic where terrestrial biomass is low, we assume that the contribution of terrestrial-derived campesterol and  $\beta$ -sitosterol is minimal compared to that produced in the ocean."

The reviewer brought up an important concern in our statements of sterol origins, which we realize is likely due to a lack a specificity on our part. In other regions of the Arctic, such as the Kara and Laptev Seas, we are aware that there can indeed be a significant terrigenous component of sterols to marine sediment, particularly in the form of campesterol and ß-sitosterol. However, we believe the confusion here relates to the fact that we initially referred to the "Arctic" in general. In the revised manuscript, we will be more specific and state in L113 that "In the Canadian Arctic Archipelago where terrestrial biomass is low (Gould et al., 2003), we assume that the contribution of terrestrial-derived campesterol and ß-sitosterol is minimal compared to that produced in the ocean". In addition to the low terrestrial biomass, there are also no rivers that drain into Baffin Bay that are comparable to those that drain the Siberian Arctic for instance. Hence, the combination of the Canadian Archipelago's low terrestrial biomass and river network, the fact that all sterols are strongly correlated (Fig. 5), and the overall marine signature of the sediment as indicated by bulk geochemistry (Fig. 3) all strongly suggest that at least in northern Baffin Bay, the 4 sterols we analyzed are all dominantly synthesized by marine organisms.

**References:**

Belt, S.T., Brown, T.A., Navarro Rodrigues, A., Cabedo Sanz, P., Tonkin, A., and Ingle, R.: A reproducible method for the extraction, identification and quantification of the Arctic sea ice proxy IP25 from marine sediments: Anal. Meth., 4, 705-713, 2012.

Cabedo Sanz, P., Smik, L., and Belt, S.T.: On the stability of various highly branched isoprenoids (HBI) lipids in stored sediments and sediment extracts: Org. Geochem., 97, 74-77, 2016.

Gould, W.A., Raynolds, M., and Walker, D.A.: Vegetation, plant biomass, and net primary productivity patterns in the Canadian Arctic: J. Geophys. Res. Atmos., 108, 1-14, 2003.

Wang, K. J., Huang, Y., Majaneva, M., Belt, S. T., Liao, S., Novak, J., Kartzinel, T. R., Herbert, T. D., Richter, N., and Cabedo-Sanz, P.: Group 2i Isochrysidales produce characteristic alkenones reflecting sea ice distribution. Nat. Comm., 20, 1-10, 2021.